# Histone methyltransferase DOT1L differentially affects the development of dendritic cell subsets

Rianne G Bouma[1,2,3,*], Willem-Jan de Leeuw[4,*], Aru Z Wang[1,2,3], Muddassir Malik[4], Joeke GC Stolwijk[1,2,3], Veronique AL Konijn[1,2,3], Anne Mensink[5], Natalie Proost[6], Maarten K Nijen Twilhaar[1,2,3], Tibor van Welsem[4], Negisa Seyed Toutounchi[1,2,3], Alsya J Affandi[1,2,3], Jip T van Dinter[7], Fred van Leeuwen[4,8], Joke MM den Haan[1,2,3]

Dendritic cells (DCs) orchestrate immune responses. Their development is controlled by transcription factors, but epigenetic mechanisms remain poorly understood. DOT1L emerges as a key epigenetic regulator in immune cells. Mapping DOT1L-mediated histone H3K79 methylation in canonical DC subsets revealed that DOT1L modified common and DC subset–specific genes. Deletion of *Dot1l* in vivo or in vitro decreased myeloid progenitors and increased cDC2s, whereas cDC1s remained unchanged. In addition, in vitro deletion of *Dot1l* led to loss of plasmacytoid DCs (pDCs) and of IFNα production upon stimulation. Upon in vivo deletion, a decrease in pDCs was only observed after subsequent in vitro expansion. This difference was likely related to insufficient replication-mediated loss of H3K79 methylation in vivo within the time frame studied. Transcriptomes of *Dot1l*-KO DC subsets exhibited enrichment of antigen presentation pathways, and MHC class II surface levels were up-regulated in pDCs. Mechanistically, inhibition of DOT1L linked the observed effects to its methyltransferase activity. Together, our data indicate that in DCs DOT1L differentially affects the development of canonical subsets and suppresses antigen presentation pathways.

## Introduction

Through their ability to integrate the innate and adaptive immune system, dendritic cells (DCs) serve as key regulators of the adaptive immune response (1, 2). DCs are commonly divided into conventional DCs (cDCs), including cDC1s and cDC2s, and plasmacytoid DCs (pDCs) (1, 2, 3). cDC1s have been identified as important stimulators of CD8[+] T cells and T helper 1 cells (Th1), whereas cDC2s have a more prominent role in the activation of Th2, Th17, and B cell responses (1, 4). The cDC2 subset is further subdivided into a Notch2-dependent T-bet[+] cDC2A subset that can secrete IL-23 and contribute to germinal center formation and antibody production (5, 6, 7, 8) and a KLF4-dependent cDC2B subset that induces Th2 responses to parasites and allergens (6, 9). pDCs are mostly involved in the response to viral infections and contribute to systemic autoimmunity (3, 10, 11). They express Toll-like receptors (TLRs) TLR7 and TLR9 and respond to single-stranded RNA or CpG with rapid production of type I interferons (IFNs) (10, 12).

The development of cDCs and pDCs in the bone marrow (BM) shows some overlapping and distinct features (1, 2, 3). Both subsets are dependent on the growth factor FMS-like tyrosine kinase 3 ligand (FLT3L). cDCs develop from the myeloid lineage via common myeloid progenitors (CMPs), monocyte dendritic cell progenitors (MDPs), and common dendritic cell precursors (CDPs). CDPs consist of multiple predisposed subpopulations (13, 14, 15) from which pre-cDC1s, pre-cDC2As, and pre-cDC2Bs differentiate. These cells migrate to the peripheral tissues and further develop into cDC1s, cDC2As, and cDC2Bs (1, 16). cDC and pDC ontogeny is a field of ongoing investigation and active debate. A recently discovered cDC subset referred to as pDC-like or transitional DCs expresses pDC-associated markers and has recently been proposed as precursors of cDC2As (17, 18, 19, 20). In addition, two developmental routes have been put forward for pDCs. The shared dependence on FLT3L and the identification of shared precursors indicated that pDCs develop from the myeloid lineage similar to cDCs (21, 22, 23). However, recently a lymphoid lineage origin for pDCs was proposed in which they develop from common lymphoid precursors (CLPs) (24, 25, 26, 27). Finally, the DC3 subset is the most recent identified DC subtype and constitutes an independent lineage that derives from Ly6C[+] MDPs (28).

Although transcriptional regulation of DC differentiation has been studied extensively, the role of the epigenome in DC differentiation is still poorly understood (29, 30, 31). At the level of

[1]Department of Molecular Cell Biology and Immunology, Amsterdam UMC Location Vrije Universiteit Amsterdam, Amsterdam, the Netherlands   [2]Amsterdam Institute for Immunology and Infectious Diseases, Cancer Immunology, Amsterdam, the Netherlands   [3]Cancer Center Amsterdam, Cancer Biology and Immunology, Amsterdam, the Netherlands   [4]Division of Gene Regulation, Netherlands Cancer Institute, Amsterdam, the Netherlands   [5]Center for Medical Genetics, Antwerp University Hospital, Edegem, Belgium   [6]Division of Experimental Animal Pathology, Netherlands Cancer Institute, Amsterdam, the Netherlands   [7]Princess Máxima Center for Pediatric Oncology, Utrecht, the Netherlands   [8]Department of Medical Biology, Amsterdam UMC, University of Amsterdam, Amsterdam, the Netherlands

Correspondence: fred.v.leeuwen@nki.nl; j.denhaan@amsterdamumc.nl
*Rianne G Bouma and Willem-Jan de Leeuw contributed equally to this work

genome organization, a role in cDC1 development and function was found for DNA methylation and the cohesion complex (32 *Preprint*, 33). At the level of the nucleosome, deficiency of the histone lysine demethylase KDM5C stimulates the development of cDC1s and cDC2Bs, and leads to increased numbers of non–IFN-producing pDCs with increased inflammatory gene expression (34). In addition, deficiency of the histone deacetylases HDAC1 and HDAC3 impairs pDC and cDC2 development, whereas cDC1 development is not affected (35, 36, 37). However, HDAC1 and HDAC3 are enzymes with broad substrate specificities (38) and may therefore affect DCs by mechanisms beyond erasing acetyl marks from chromatin. In thymocytes, HDAC1 has been shown to crosstalk with the chromatin modifier disruptor of telomeric silencing 1–like (DOT1L) (39). This histone methyltransferase is emerging as an important gatekeeper in the differentiation and functioning of macrophages, NK, B, and T cells (39, 40, 41, 42, 43, 44, 45, 46, 47, 48, 49). Recent studies suggest that DOT1L affects DC maturation and tolerance induction, although its role in DC subset differentiation remains unknown (31, 50, 51).

DOT1L has one canonical substrate and is the sole methyltransferase of histone 3 lysine 79 (H3K79) (52, 53, 54). H3K79 methylation is predominantly deposited in the promoter-proximal gene-body regions of transcribed genes (55). DOT1L facilitates the mono (me1)-, di (me2)-, and trimethylation (me3) of H3K79, and its activity is enhanced through transcriptional elongation and transcription-associated histone modifications (53, 56). In agreement with this cotranscriptional regulation, the level of H3K79 methylation (H3K79me) correlates with the level of gene expression (53, 54, 57, 58). Highly specific inhibitors have been developed that specifically block DOT1L catalytic activity, lead to loss of H3K79me, and have potential therapeutic applications (59, 60, 61, 62, 63, 64). However, H3K79 methylation is remarkably stable and mainly lost through cell division (49, 63, 64), which is in agreement with the absence of a demethylase known to specifically act on this mark (42, 44). Although DOT1L activity mainly seems restricted toward H3K79me, cellular functions of DOT1L independent of its methyltransferase activity have been described as well (65, 66, 67, 68, 69, 70).

Here, we characterized the role of DOT1L in the differentiation of DCs both in vitro and in vivo. We show that genetic ablation of *Dot1l* followed by in vitro cell culture of BM cells led to a decrease in myeloid precursors and pDCs, and an increase in cDC2s. This role of DOT1L is linked to its catalytic activity, as chemical inhibition recapitulated these findings. In classical DC subsets, we confirmed that dimethylation of H3K79 (H3K79me2) is deposited in a canonical pattern across gene bodies of transcribed genes. The targets of DOT1L include genes encoding transcription factors involved in pDC differentiation (*Tcf4*, *Bcl11a*, *Irf8*, and *SpiB*), and the expression of these transcription factors was decreased in *Dot1l*-KO pDCs in vivo (71, 72, 73, 74, 75, 76, 77). In addition, antigen presentation–related pathways were enriched in the transcriptomes of all *Dot1l*-KO DC subsets. Combined, these results suggest that DOT1L plays an essential role in the function and differentiation of DCs, especially in pDCs. Our quantitative H3K79me2 epigenomics map, combined with perturbation and transcriptomic data, provides an important resource for gaining

more insights into how DC development and function are guided by epigenetic modulation.

## Results

### Dot1l expression and H3K79me2 methylation state in DC subsets

To explore the potential importance of DOT1L in the development of DCs, we first investigated the presence of *Dot1l* mRNA in DCs (precursors) through publicly available RNA-sequencing datasets. A scRNA-sequencing dataset from Schlitzer et al showed that DC progenitors from the BM indeed express *Dot1l* (Fig 1A) (13). Moreover, bulk RNA-sequencing data from sorted DCs in the spleen from Rodrigues et al revealed the presence of *Dot1l* in cDC1s, cDC2s, and pDCs, although with some variation between samples (Fig 1B) (17). Lastly, these findings are in line with bulk RNAseq data from Yoshida et al, which confirmed that *Dot1l* expression in DCs and their precursors clustered around the median of all immune subsets analyzed (Fig S1A) (78).

Because *Dot1l* appeared to be expressed in several DC subtypes, we sorted pDCs (Siglec-H$^{pos}$ BST2$^{pos}$), cDC1s (CD11c$^{pos}$ XCR1$^{pos}$), and cDC2s (CD11c$^{pos}$ Sirpα$^{pos}$) from WT C57BL/6 mice and performed ChIPseq for H3K79me2, as well as RNAseq. Although DOT1L is responsible for depositing H3K79me1 and H3K79me2 in transcribed regions, H3K79me2 shows a stronger correlation with gene expression and has been studied more extensively. To quantitatively compare H3K79me2 peaks and levels between the DC populations, a yeast chromatin spike-in was added to all samples in parallel and co-immunoprecipitated as recently described (Fig 1C) (49). Within actively transcribed genes, H3K79me2 was predominantly deposited between the transcriptional start site (TSS) and the first internal exon (FIE), consistent with previous findings in other cell types (Fig 1D) (40, 42, 55). Comparison of the spike-in normalized peaks revealed that global H3K79 methylation levels were comparable between the sorted DC populations (Fig S1B), indicating that DOT1L has similar activity in the different DC subsets.

Analysis of the methylated regions revealed that some genes had similar H3K79me2 levels in all subtypes (e.g., *Runx1*, Fig 1E, top), whereas others were only methylated in one DC population (e.g., *Tcf4*, Fig 1E, bottom). Overall, 4,522 H3K79me2 peaks were identified as shared between all sorted cell populations, whereas 2,161 peaks uniquely mapped to pDC, 3,096 to cDC1, and 2,647 to the cDC2 population (Figs 1F and S1C). GO analyses revealed that the pDC-specific peaks were consistent with gene signatures from previous pDC-related publications, suggesting that *Dot1l* might indeed be involved in the regulation of key genes within this cell population (Fig S1D, Table S1).

As H3K79me2 methylation typically correlates with active transcription (53, 54, 57, 79), we performed RNAseq to confirm whether differences in H3K79me2 deposition between the DC populations were related to transcriptional differences. To compare the ChIPseq and RNAseq data, we quantified the H3K79me2 signal near the TSS (Fig S1E, Table S2) and correlated this to transcript length–normalized gene expression (Fragments Per Kilobase of exon per Million mapped fragments—FPKM, Table S3). Generally, the H3K79me2 signal near the TSS correlated well

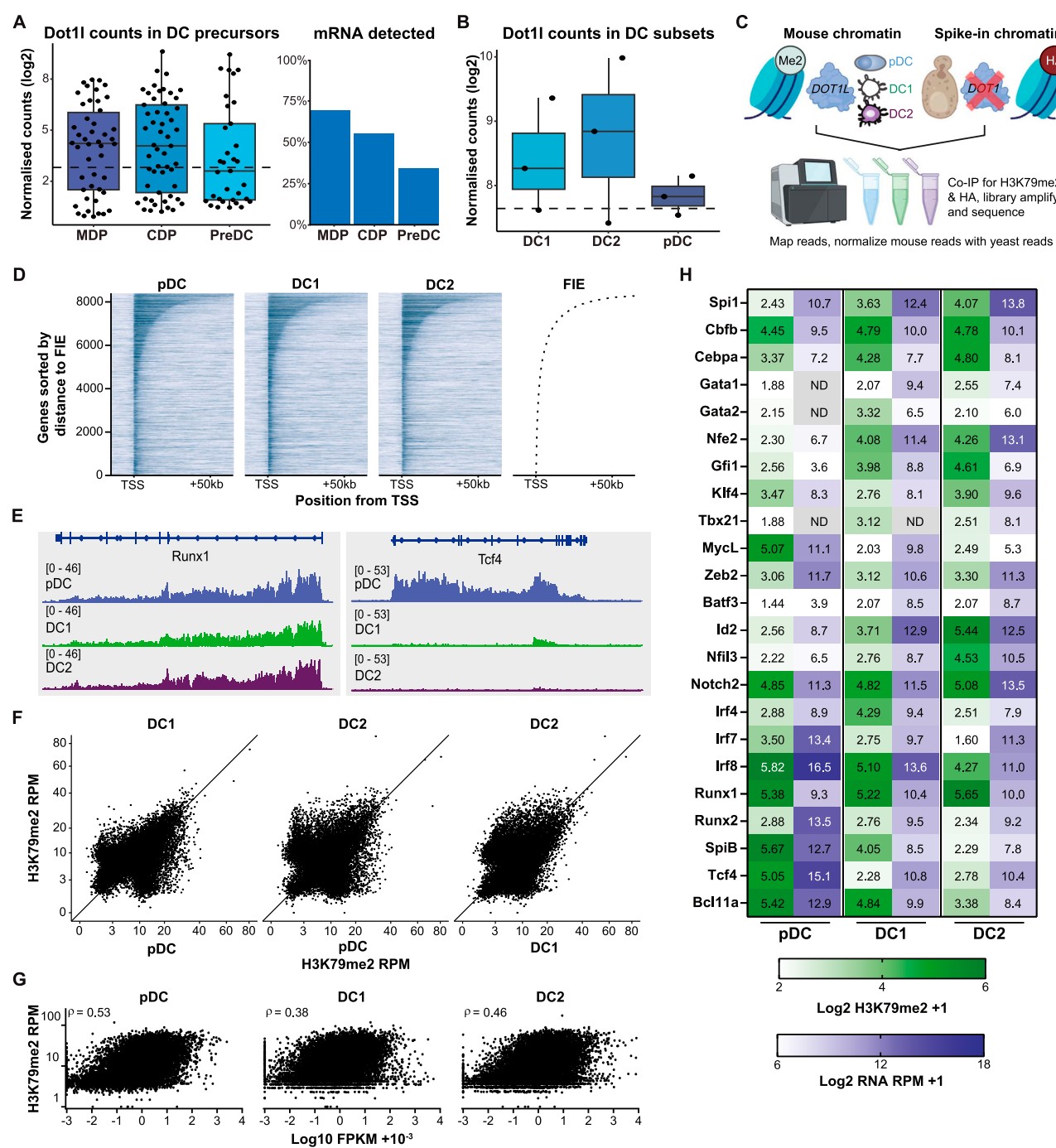

**Figure 1. DOT1L H3K79me2 methylation in DC subsets.**
**(A)** *Dot1l* counts derived from scRNA-sequencing analysis of sorted DC precursors. Left: median of all genes is plotted as a dashed line. Right: percentage of cells with *Dot1l* counts higher than 0 (*Dot1l* detected) versus cells with no *Dot1l* counts (not detected). The dataset was acquired from reference 13. **(B)** *Dot1l* counts derived from bulk RNA-sequencing analysis of sorted DC subsets. The median of all genes is plotted as a dashed line. The dataset was acquired from reference 17. **(C)** Spike-in–normalized H3K79me2 ChIPseq workflow. Sorted Siglec-H[pos] BST2[pos] pDC, CD11c[pos] XCR1[pos] cDC1, and CD11c[pos] Sirpα[pos] cDC2 pools (N = 2 biological replicates) were spiked with chromatin from a *dot1Δ*- and HHT2-HA–tagged yeast strain and co-immunoprecipitated for H3K79me2 and HA in parallel. Reads mapped to the *S. cerevisiae* genome were used to normalize the H3K79me2 reads within each sample. **(D)** Tornadoplot showing the H3K79me2 signal in relation to the transcription start site (TSS) and first internal exon (FIE). For each subset, the top 50% active genes based on expression are displayed, ranked according to distance to the FIE (short to long), as displayed in the rightmost graph. **(C, E)** Snapshot of the normalized H3K79me2 data obtained from (C) for two representative genes. Data were visualized using *IGV*. **(F)** Dot plots comparing H3K79me2 peaks between the sorted DC subsets. **(G)** Dot plot correlating the H3K79me2 signal near the TSS (N = 2) and RNA expression (N = 1) for each sorted DC subset. Expression data were normalized for transcript length by calculating the Fragments Per Kilobase of transcript per Million mapped reads (FPKM) values for each gene. The calculated Spearman's rank correlation coefficients are displayed within each graph. **(H)** Heatmap of selected transcription factors, quantifying the H3K79me2 signal near the TSS (green gradient) and expression based on RNAseq (blue gradient) in the sorted subsets. ND, not detected.

between biological replicates (Fig S1F) and moderately with transcription. The correlation was strongest in pDCs ($\rho$ = 0.53), followed by cDC2s ($\rho$ = 0.46), and weakest in the cDC1 ($\rho$ = 0.38) population (Fig 1G). For some genes encoding transcription factors that regulate DC differentiation and function (*Tcf4*, *Bcl11a*, *Irf8*, and *SpiB*), high levels of both H3K79me2 and mRNA expression were observed in the sorted pDCs, suggesting that H3K79me2 was linked to transcriptional activity (Fig 1H). In conclusion, the normalized ChIPseq data indicate that DOT1L induces similar levels of global H3K79me2 in DC subsets but they also show specific differences in specific genes in certain DC subsets. As these changes are, in part, accompanied by changes in expression, this suggests that DOT1L-mediated H3K79me2 might be involved in directly or indirectly regulating DC differentiation, especially for the pDC subset.

### Decreased development of pDCs and increased development of cDC2s upon in vitro deletion of Dot1l in FLT3L- and SCF-supplemented BM cultures

Considering that the ChIPseq revealed that *Dot1l* is active in DCs and methylated both common and unique genes, we next investigated what role *Dot1l* plays in DC development using an in vitro FLT3L/SCF BM DC culture system (80, 81, 82). We used a mouse strain (Cre-ER^T2;*Dot1l*^fl/fl) in which the genetic ablation of *Dot1l* (KO; loss of exon 2) can be induced through the addition of 4-hydroxytamoxifen (4-OHT) (Fig 2A). It should be noted that after 4-OHT–induced *Dot1l* KO, the loss of DOT1L-associated H3K79 methylation is delayed as it primarily depends on dilution by cell division (83). FLT3L/SCF-supplemented Cre-ER^T2;*Dot1l*^wt/wt (WT) BM culture led to the generation of BST2^pos Siglec-H^pos pDCs and CD11c^pos MHC class II^pos Sirpα^pos cDC2s, as well as CD11c^pos MHC class II^pos XCR1^pos cDC1s on day 7 (Figs 2B and C and S2). All subsets were pregated as Live CD45^pos Lineage^neg cells. Loss of *Dot1l* through in vitro treatment with 4-OHT decreased the number and frequency of pDCs from 15.4% to 4.2% on day 7 after the start of the culture (percentage of Live CD45^pos Lineage^neg cells; Figs 2C and D and S2). In contrast, the number and frequency of cDC2s increased from 9.7% to 16.9% in *Dot1l*-KO cultures, whereas cDC1s were unaffected (Figs 2C and D and S2). In conclusion, KO of *Dot1l* in total murine BM decreased the number of pDCs and increased the number of cDC2, after culturing for 7 d with FLT3L and SCF.

### In vivo deletion of Dot1l in the bone marrow followed by in vitro expansion leads to reduced pDCs and myeloid precursors

Deletion of *Dot1l* in vitro appeared to affect the development of DCs in BM cultures. Although DOT1L has a short half-life of ~2 h (84, 85), H3K79me is a stable histone modification and loss of the epigenetic mark upon inactivation of DOT1L has been reported to be mainly dependent on dilution by cell division (83, 86). Proliferation during the 7-d BM cultures likely facilitated the loss of H3K79me after the loss of DOT1L in an early stage of the cultures. To explore whether in vivo loss of DOT1L protein led to a direct effect on DC (precursor) frequency, we treated the Cre-ER^T2;*Dot1l*^fl/fl and Cre-ER^T2;*Dot1l*^wt/wt mice in vivo with tamoxifen to generate *Dot1l*-deficient mice. The mice were injected intraperitoneally with

tamoxifen on three consecutive days after which the genetic ablation of *Dot1l* was confirmed by PCR (Fig S3A–C). On day 3, cells isolated from the spleen and BM were stained with an antibody panel to identify DC precursors and DC subsets. The identification of myeloid and lymphoid precursors in the BM was based on the presence or absence of previously described markers on these cell subsets (Fig S3A) (1, 22, 87, 88, 89, 90, 91). All subsets were pregated as Live Lineage^neg Autofluorescence (AF)^neg cells. Subsequently, CMPs, MDPs, and CDPs were gated from MHC class II^neg CD11c^neg CD135^pos cells and defined as CD117^hi CD115^neg CMPs, CD117^int CD115^hi MDPs, and CD117^neg CD115^hi CDPs. CLPs were defined as MHC class II^neg CD11c^neg CD135^neg CD127^pos; pre-pDCs were gated as CD117^neg CD135^pos CD11c^pos Siglec-H^pos Ly-6D^pos, pDCs as B220^pos Siglec-H^pos BST2^pos, and cDC1s and cDC2s as B220^neg MHC class II^hi CD11c^hi Sirpα^neg/pos (Fig S3A). On day 3, we did not observe changes in precursor or DC populations in the BM (Fig S3D) or the spleen (Fig S3E and F). These results indicate that loss of DOT1L protein itself does not result in direct changes in DC subsets and precursors and suggest that more extensive proliferation is necessary to lose H3K79me. Moreover, if the effect of DOT1L is at the level of bone marrow precursors, then effects on DC subsets will be delayed as differentiation of DC subsets will require 5–7 d (92).

To recapitulate our previous results and investigate changes in DC precursors in a BM culture setting, isolated BM cells from tamoxifen-treated Cre-ER^T2;*Dot1l*^fl/fl and Cre-ER^T2;*Dot1l*^wt/wt mice were cultured with FLT3L and SCF as described above (Fig 3A). On day 7, the cultures were harvested and stained to identify precursors and DC subsets (Fig S3A). In line with the previous in vitro BM cultures, pre-pDCs and pDCs showed a clear reduction, whereas cDC2s were increased upon *Dot1l* KO (Fig 3B). A clear reduction in the percentage of cells belonging to the myeloid lineage was observed, whereas the lymphoid compartment and the cDC1 subset remained unaffected in the *Dot1l*-KO cultures (Fig 3B). CMPs, MDPs, and CDPs were all significantly reduced after *Dot1l* KO in contrast to CLPs, which remained unaffected. The discrepancy in abundance between DC precursors, such as CDP, which were decreased, and cDC1 and cDC2s, which were stable or were increased, suggests that *Dot1l*-KO DCs might undergo an accelerated differentiation trajectory toward these more terminally differentiated cell types.

pDCs have been described to respond to class A CpGs by rapidly producing large amounts of type I IFNs (10, 12). A recent study in addition observed type I IFN production by pre-cDC2s after class A CpG stimulation (93). To determine whether the remaining DCs from *Dot1l*-deficient BM cultures were still functional, BM cells were stimulated with class A CpGs (ODN 1585) overnight on day 7 and IFNα levels in the supernatant were detected using ELISA. This revealed that IFNα production after CpG stimulation was absent in the *Dot1l*-KO cultures (Fig 3C), which correlates with the decreased number of pDCs. However, the remaining *Dot1l*-KO DCs still up-regulated the maturation marker CD86 in response to class A CpGs (Fig 3D). To summarize, in vivo deletion of *Dot1l* did not result in changes in immune populations in the BM and spleen on day 3 after the first tamoxifen injection. However, a subsequent 7-d culture of the BM cells resulted in a reduction of myeloid precursors, pre-pDCs, and pDCs and up-regulation of cDC2s. This indicates that loss of DOT1L alone is not sufficient to observe the

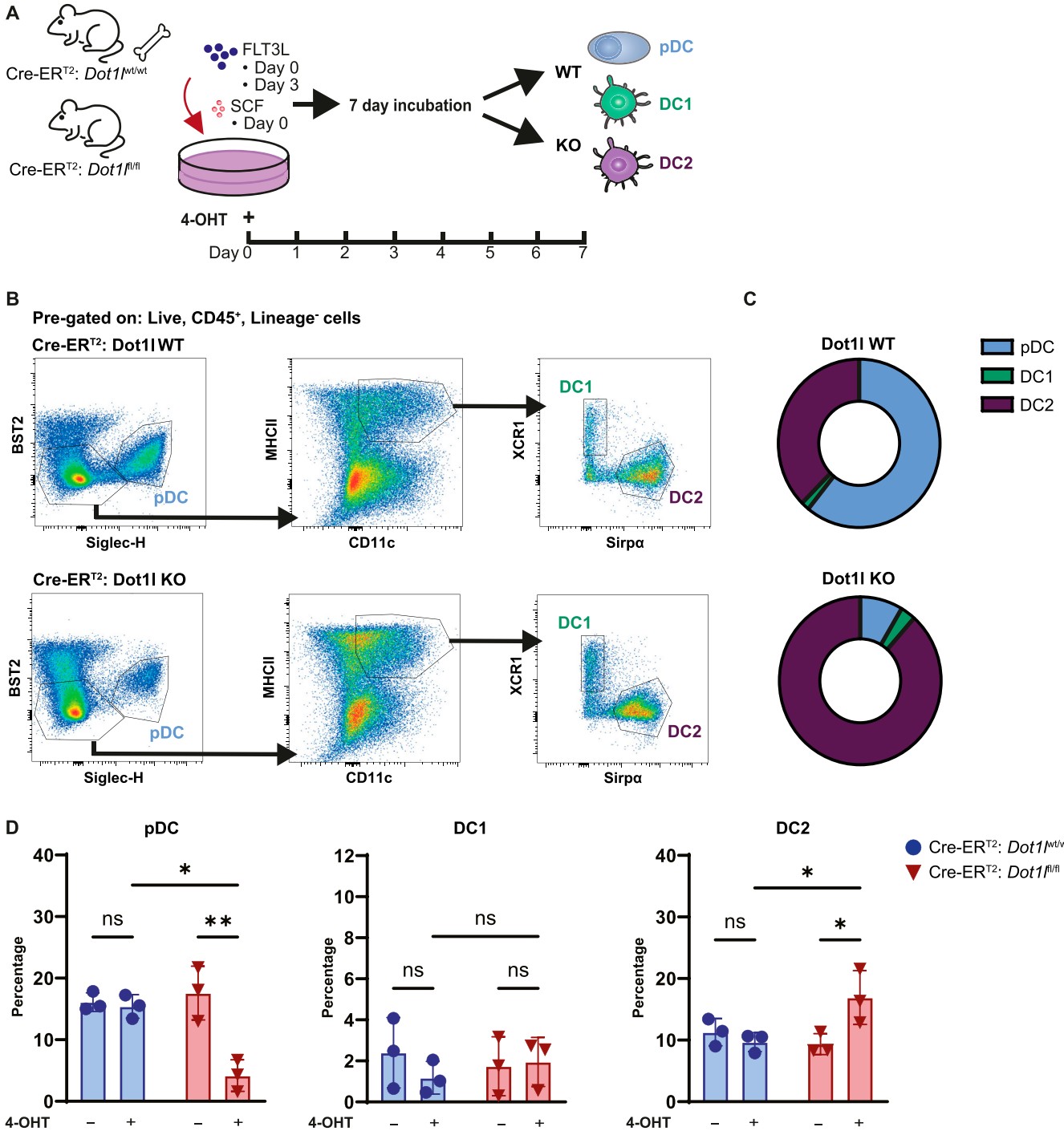

**Figure 2. Decreased development of pDCs and increased development of cDC2s upon in vitro deletion of *Dot1l* in FLT3L/SCF BM cultures.**
**(A)** Schematic overview of the setup of the 7-d BM culture. Total BM of Cre-ER[T2] *Dot1l*[wt/wt] and Cre-ER[T2] *Dot1l*[fl/fl] mice was cultured with or without 4-hydroxytamoxifen and with SCF and FLT3L to facilitate the development of DCs. **(B)** Overview of the gating strategy on total SCF and FLT3L BM cultures on day 7. The top row is representative of a *Dot1l*WT sample, and the bottom row is representative of a *Dot1l*-KO sample. **(C)** DC subset percentages of Live CD45[pos] Lineage[neg] cells. Circle plots represent *Dot1l* WT (Cre-ER[T2];*Dot1l*[wt/wt] + 4-hydroxytamoxifen) and *Dot1l*-KO (Cre-ER[T2];*Dot1l*[fl/fl] + 4-hydroxytamoxifen) conditions. **(D)** Comparison of DC subset percentages of Live CD45[pos] Lineage[neg] cells in Cre-ER[T2] *Dot1l*[wt/wt] and Cre-ER[T2] *Dot1l*[fl/fl] conditions, with and without 4-hydroxytamoxifen to induce KO of *Dot1l*. The individual dots represent the average of technical replicates in one independent experiment, with a total of three independent experiments performed. Error bars indicate the mean ± SD. *P < 0.05, **P < 0.01, ***P < 0.001, ****P < 0.0001.

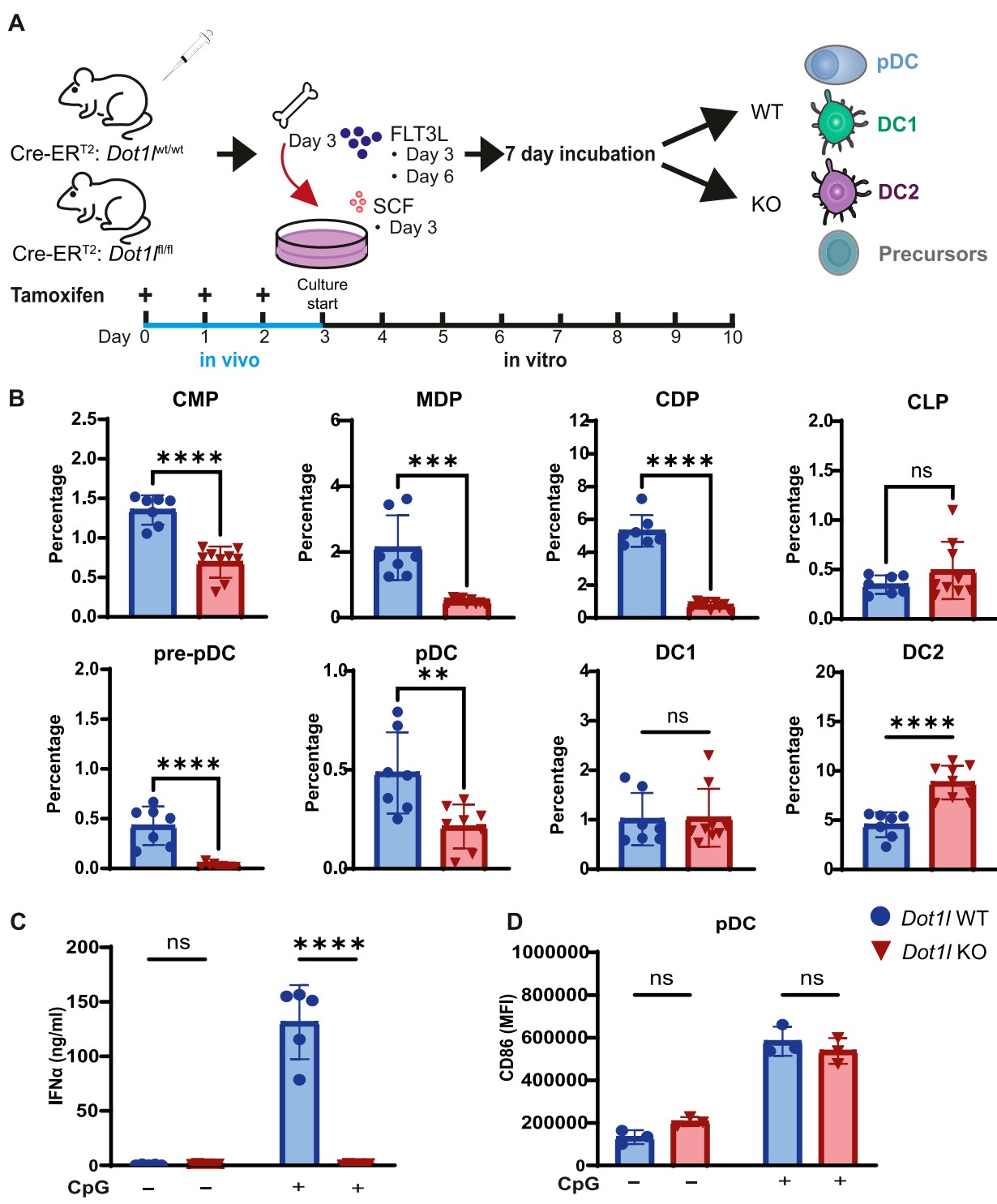

**Figure 3. Reduced frequency of pDCs, and myeloid, but not lymphoid, precursors after in vivo deletion of *Dot1l* and subsequent 7-d SCF and FLT3L BM culture.**
**(A)** Cre-ER^T2 *Dot1l*^wt/wt and Cre-ER^T2 *Dot1l*^fl/fl mice were injected three times with tamoxifen. Subsequently, BM was cultured with SCF and FLT3L. Seven days after the start of the culture, DC and precursor subset composition was evaluated. **(B)** Comparison of myeloid and lymphoid precursor percentages of Live Lineage^neg AF^neg cells in *Dot1l*-WT (Cre-ER^T2 *Dot1l*^wt/wt) and *Dot1l*-KO (Cre-ER^T2 *Dot1l*^fl/fl) conditions. CMPs, MDPs, and CDPs were gated from MHC class II^neg CD11c^neg CD135^pos cells and defined as CD117^hi CD115^neg CMPs, CD117^int CD115^hi MDPs, and CD117^neg CD115^hi CDPs. CLPs were defined as MHC class II^neg CD11c^neg CD135^neg CD127^pos. Pre-pDCs were gated as CD117^neg CD135^pos CD11c^pos Siglec-H^pos Ly-6D^pos, pDCs as B220^pos Siglec-H^pos BST2^pos, and cDC1s and cDC2s as B220^neg MHC class II^hi CD11c^hi Sirpα^neg/pos. **(C)** On day 7 after the start of the culture, BM cells were stimulated with class A CpGs (ODN1585) overnight. IFNα levels were determined using ELISA. **(D)** After class A CpG stimulation, cells were stained for maturation markers. Conditions are shown with and without the addition of CpGs. The individual dots represent individual mice with a total of three independent experiments performed. Error bars indicate the mean ± SD. *$P < 0.05$, **$P < 0.01$, ***$P < 0.001$, ****$P < 0.0001$.

changes in DC subsets and that proliferation and/or differentiation is necessary.

### In vitro inhibition of DOT1L in WT BM cultures results in decreased generation of pDCs and increased generation of cDC2s

DOT1L is the sole methyltransferase of H3K79 but has also been reported to have cellular roles independent of H3K79 methylation (65, 66, 67, 68, 69, 94). To investigate the role of the catalytic activity of DOT1L in DC differentiation, we treated the cells in our culture system with the potent and selective DOT1L inhibitor SGC-0946 (Fig 4A) (59). Treatment with SGC-0946 did not affect the viable cell count of the FLT3L/SCF BM cultures (Fig S4). Similar to the *Dot1l*-KO cultures, we observed a decrease in CDPs, pre-pDCs, and pDCs after 7 d of culture in the presence of SGC-0946. Furthermore, cDC2s were increased and the fraction of CLPs was unaffected. However, no changes were observed in CMP and MDP populations (Fig 4B). In line with our previous results, the ability of CpG-stimulated DCs to produce IFNα was completely abolished after DOT1L inhibition, whereas CD86 expression did not change significantly (Fig 4C). To summarize, inhibition of the methyltransferase activity of DOT1L resulted in similar effects as *Dot1l* KO. Because DOT1L inhibitor treatment does not affect the stability of the DOT1L protein (95), this indicates that the phenotype induced by DOT1L is mainly due to its catalytic activity.

### In vivo deletion of Dot1l affects specific myeloid and lymphoid precursor subsets

Although the in vivo genetic ablation of *Dot1l* in Cre-ER$^{T2}$;*Dot1l*$^{fl/fl}$ mice did not result in changes in the DC (precursor) populations 3 d after the first treatment with tamoxifen, the subsequent 7-d in vitro culture altered DC precursor and DC subset frequencies. Combined with the DOT1L inhibitor experiments, this supports the hypothesis that the effect of DOT1L on DC differentiation is associated with its methyltransferase activity. Earlier studies estimated the half-life of DCs to be 5–7 d (92). Therefore, to accommodate for both the time needed to passively dilute H3K79me2 and subsequent differentiation, we set up an in vivo experiment in which we evaluated the cell populations in the spleen and BM on day 12 after the start of tamoxifen administration (Fig 5A). *Dot1l* deletion efficiency was more than 90% in both BM and spleen (Fig S5A), whereas staining for H3K79me2 revealed that ~80% of BM cells and 10% of splenocytes showed reduced H3K79me2 levels (Fig S5B and C). Unfortunately, it was not possible to determine H3K79me2 staining in specific cell subsets, as the harsh denaturation conditions required for this staining could not be combined with the elaborate panel of surface markers needed for the identification of these cells.

On day 12 after the first dose of tamoxifen, we observed a decrease in CMPs, MDPs, and CDPs, and an increase in CLPs and cDC2s in the BM, similar to the in vitro BM cultures (Fig 5B). However, no significant difference was observed in the frequency of pre-pDC and pDC populations (Fig 5B). In the spleen, no differences in the frequencies of cDC1, cDC2, and pDC subsets were observed (Fig S5D). To investigate whether loss of *Dot1l* resulted in proliferative changes, we stained for Ki67 in BM and splenic DC

subsets and precursors. Interestingly, MDP, CDP, and pre-pDC in the BM and DC subsets and monocytes in the spleen showed increased Ki67 staining, whereas CLP did not exhibit increased Ki67 expression (Fig S5E). These data indicate that loss of *Dot1l* did not directly impair cell proliferation of DC subsets. In addition, an inverse relation between cell numbers and Ki67 staining was observed, such as high CLP numbers with no change in Ki67 and decreased numbers of CDP with high Ki67 staining, suggesting potentially increased proliferation in response to decreased differentiation.

Flow cytometry analyses of bone marrow cells revealed a significant increase in MHC class II expression on pDCs and to a lesser extent in cDC2s and a decrease in B220 expression on pDCs in *Dot1l*-KO mice compared with *Dot1l*-WT mice (Fig 5C). BM *Dot1l*-KO pDCs retained the ability to up-regulate CD86 and produce type I IFNs after stimulation with class A CpGs as detected by intracellular IFNαβ staining (Fig 5D). In the spleen, a strong increase in CD11b expression on pDCs, cDC1s, and cDC2s, as well as an increase in Ly6C expression on pDCs, was detected in *Dot1l*-KO mice (Fig S5F). After overnight stimulation of splenocytes with class A CpGs, *Dot1l*-KO pDCs still produced type I IFNs and up-regulated maturation marker CD86 (Fig S5G).

In summary, after 12 d of *Dot1l* deletion in vivo we detected changes in DC precursor and cDC2 frequencies in the BM, but not in the spleen. In addition, phenotypic changes were observed in DC subsets from both organs. In contrast to the in vitro experiments, pDCs were not reduced in *Dot1l*-KO mice and retained their ability to produce type I IFNs and to up-regulate maturation marker CD86.

To gain further insight into the consequences of *Dot1l* loss in vivo, we performed RNAseq on sorted WT and KO populations (Siglec-H$^{pos}$ BST2$^{pos}$ pDC, CD11c$^{pos}$ XCR1$^{pos}$ cDC1, and CD11c$^{pos}$ Sirpα$^{pos}$ cDC2) from the BM at day 12 after the start of tamoxifen treatment (see Table S3). Principal component analysis (PCA) revealed that KO cells clustered closest to their WT counterparts, suggesting that the global changes upon *Dot1l* loss were modest (Fig S5H). Overall, 228 genes were down-regulated in expression in all KO cell types (Log$_2$FC < −0.5), and based on GO analysis, these genes were mostly related to metabolic pathways and the regulation of transcription (Figs 5E and S5I and Table S4). In contrast, genes that were down-regulated in sorted *Dot1l*-KO pDC (452 genes based on Log$_2$FC < −1.0) were enriched for several pathways that negatively regulate (myeloid) cell differentiation, perhaps underlying the loss of pDC identity as previously observed upon subsequent cell culture in vitro (Fig 5F). In line with this, several transcription factors involved in pDC differentiation (e.g., *Runx2*, *Tcf4*, *Bcl11a*, *Irf8*, and *SpiB*) were modestly down-regulated in *Dot1l*-KO pDC, whereas changes for the remaining transcription factors and cell types were more variable (Fig 5G).

Similarly, most up-regulated genes were unique in each cell type, with minimal overlap between the cell types. Nevertheless, genes up-regulated in all KO populations (319 genes based on Log$_2$FC > 0.5) enriched for various antigen presentation– and activation-related pathways (Fig 5H). This corroborates earlier results (e.g., Fig 5C) that suggest a more pro-inflammatory state in *Dot1l*-KO BM and spleen. Correlating transcriptional changes in KO with H3K79me2 signal (in WT) revealed that the genes

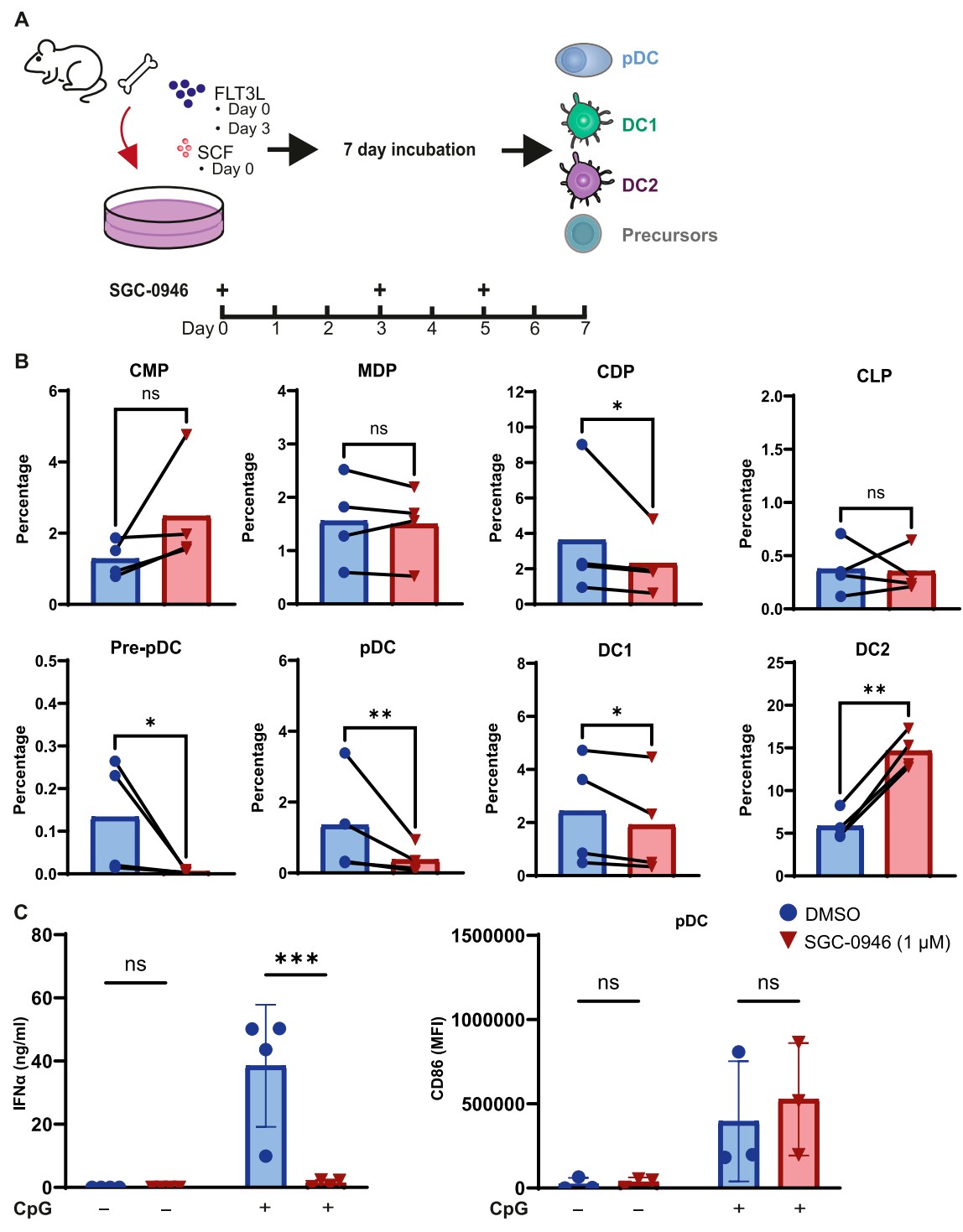

**Figure 4. In vitro inhibition of DOT1L in WT BM cultures results in decreased generation of pDC and increased generation of cDC2.**
**(A)** Schematic overview of the setup of the 7-d BM culture. Total BM of WT C57BL/6 mice was cultured with SCF and FLT3L to facilitate the development of DCs. DOT1L inhibitor SGC-0946 (1 $\mu$M) was added at days 0, 3, and 5 to ensure constant inhibition of DOT1L during DC development. DMSO (0.1%) was used as a vehicle control. **(B)** Comparison of myeloid and lymphoid precursor percentages of Live Lineage$^{neg}$ AF$^{neg}$ cells in cultures with and without DOT1L inhibitor. CMPs, MDPs, and CDPs were gated from MHC class II$^{neg}$ CD11c$^{neg}$ CD135$^{pos}$ cells and defined as CD117$^{hi}$ CD115$^{neg}$ CMPs, CD117$^{int}$ CD115$^{hi}$ MDPs, and CD117$^{neg}$ CD115$^{hi}$ CDPs. CLPs were defined as MHC class II$^{neg}$ CD11c$^{neg}$ CD135$^{neg}$ CD127$^{pos}$. Pre-pDCs were gated as CD117$^{neg}$ CD135$^{pos}$ CD11c$^{pos}$ Siglec-H$^{pos}$ Ly-6D$^{pos}$, pDCs as B220$^{pos}$ Siglec-H$^{pos}$ BST2$^{pos}$, and cDC1s and cDC2s as B220$^{neg}$ MHC class II$^{hi}$ CD11c$^{hi}$ Sirp$\alpha^{neg/pos}$. **(C)** On day 7, BM cells were stimulated with class A CpGs (ODN1585) overnight. IFN$\alpha$ levels were determined in the supernatant using ELISA, and maturation markers were evaluated by flow cytometry. Conditions are shown with and without the addition of CpGs. The individual dots represent the average of technical replicates in one independent experiment, with a total of three independent experiments performed. Error bars indicate the mean ± SD. *$P < 0.05$, **$P < 0.01$, ***$P < 0.001$, ****$P < 0.0001$.

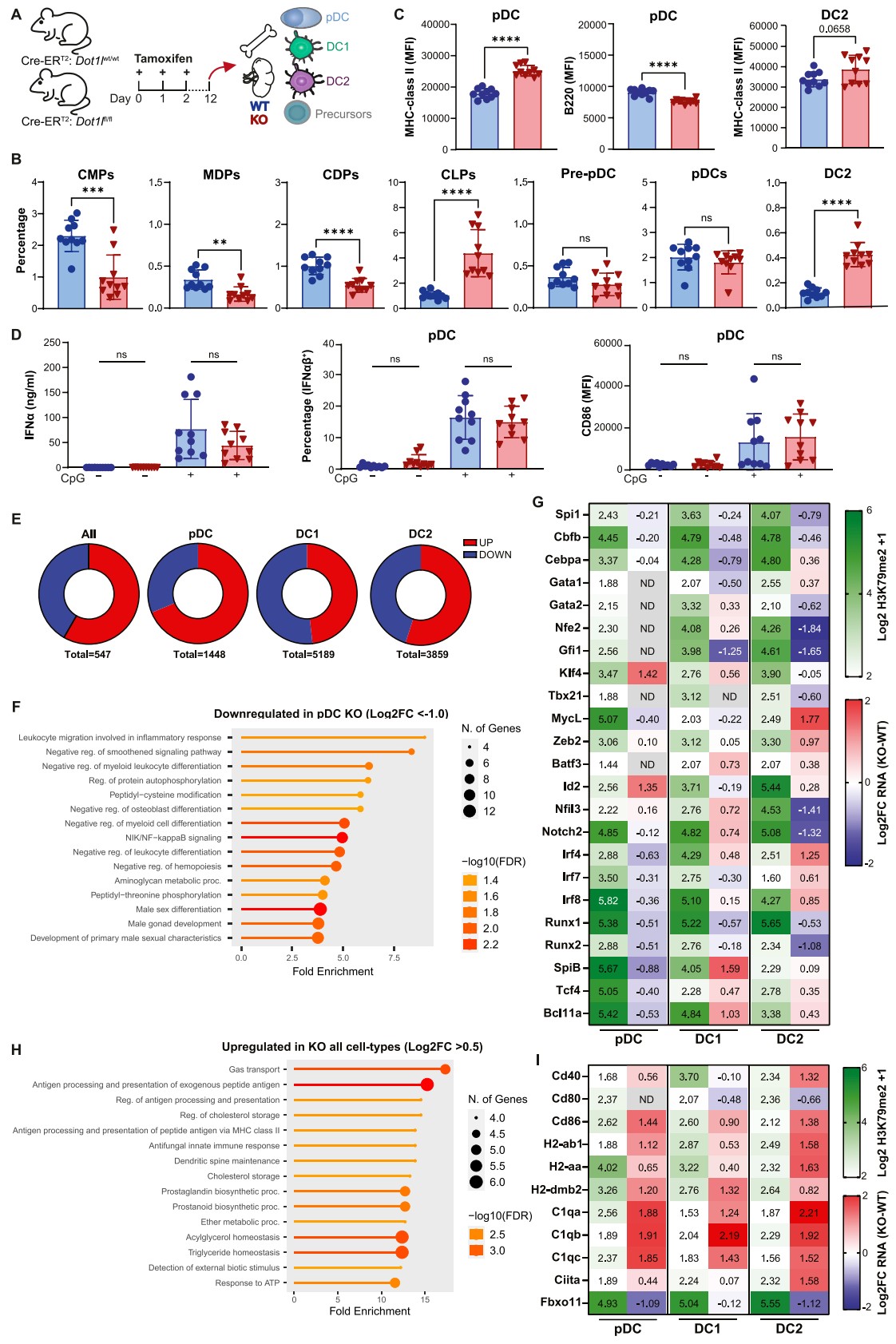

underlying these up-regulated signatures were generally hypomethylated (Fig 5I). This suggests that these genes are likely indirect targets of the loss of H3K79me2. Among the enriched activation-related genes were several activation markers (*Cd40*, *Cd80*, *Cd86*), MHC class II complex members (*H2-ab1*, *H2-aa*, *H2-dmb2*), and classical complement factors known to be expressed in macrophages (*C1qa*, *C1qb*, *C1qc*). Because macrophage blebs are known to bind to other cell types and thereby can contaminate the RNAseq data (96), we evaluated RNA expression of macrophage markers *Adgre1*, *Siglec1*, *Mertk*, and *Vcam1* in the DC subsets. The altered expression of these markers in *Dot1l KO* was not strictly correlated with the altered expression of C1qa/b/c, indicating that macrophage blebs do not solely explain the up-regulation of the classical complement factors. For example, we detected increased expression of these macrophage markers in the sorted *Dot1l*-KO pDC and DC2 subsets, whereas these macrophage genes were absent, unaltered, or only modestly affected in their expression in DC1 (see Table S3). Therefore, although macrophage contamination cannot be fully excluded, it is unlikely that the observed changes in C1q transcripts are solely due to macrophage contamination.

The expression of MHC class II genes is predominantly mediated through the master regulator CIITA and its negative regulator FBXO11 (97). *Ciita* was H3K79-hypomethylated in all cell types, whereas *Fbxo11* was hypermethylated (Fig 5I), suggesting that *Dot1l* might regulate MHC class II expression levels indirectly via *Fbxo11* (Figs 5I and S5J). *Dot1l*-KO pDCs and cDC2s showed an increase in *Ciita* expression and a decrease in *Fbxo11* expression, which might underlie the increased antigen presentation signatures predominantly observed in these two cell types (Fig 5I). Lastly, although HDACs have been linked to both DC differentiation and *Dot1l* function, we observed no consistent changes in their expression upon *Dot1l* KO, suggesting that the potential interplay between these proteins is not mediated solely through H3K79me2 levels (Fig S5K).

Combined, *Dot1l* potentially affects the differentiation of pDCs via positive regulation of a number of transcription factors, while also suppressing the basal activation state of all the sorted DC subtypes. Further studies will be required to further elucidate the mechanism underlying the regulation of these differentiation- and activation-related pathways.

# Discussion

DCs serve as key regulators of the overall immune response. Their differentiation trajectory is complex, involving both lymphoid and myeloid lineages. The role of epigenetic mechanisms guiding DC differentiation is still relatively unexplored (1, 29, 30). Recently, histone deacetylases and a demethylase have been implicated in the differentiation of various DC subtypes (34, 35, 36, 37), but otherwise, the role of epigenetic mechanisms in DC development remains understudied compared with other immune cells. Within this study, we identified a role of the histone methyltransferase DOT1L in DC differentiation toward the pDC, cDC1, and cDC2 subtypes. Previous studies have described a central role of methyltransferase DOT1L in the differentiation of other immune cell types, such as macrophages, NK cells, and B and T lymphocyte cells (40, 41, 42, 43, 45, 46, 47), and recently, DOT1L has been linked to DC function as well (50, 51). However, to our knowledge, the impact of DOT1L on the differentiation of specific DC progenitors toward DC subsets remains largely unexplored.

DOT1L is a histone methyltransferase that specifically deposits H3K79me on active genes. H3K79me2 ChIPseq on sorted DC populations revealed a distribution pattern that started around the TSS and dropped off after the first internal exon was reached, a pattern consistent with that observed in other cell types (40, 42, 49, 55). Although there was substantial overlap in H3K79me2 between the three subsets, pDCs, cDC1s, and cDC2s all contained distinct peaks as well, which suggests differential regulation of these subtypes by DOT1L. The correlation between the H3K79me2 signal around the TSS and transcription based on RNAseq was strongest in pDCs. Intriguingly, several transcription factors involved in pDC development, including *Irf8*, *SpiB*, *Tcf4*, and *Bcl11a*, were abundantly methylated in pDCs (71, 72, 73, 74, 75, 76, 77). This observation pointed to a potential role of DOT1L in the regulation of DC differentiation and especially of pDC.

When DOT1L was genetically ablated in vitro, we observed a decrease in pDC differentiation, whereas cDC2 numbers increased and cDC1s were unaltered after 7 d of culture with FLT3L and SCF. In vivo KO of *Dot1l* did not result in changes in DCs or their precursors in the BM on day 3, but a subsequent 7-d culture period revealed again changes in pDC and DC2 populations, as well as a decrease in myeloid precursors. These DC cultures were unable to produce IFNα upon class A CpG stimulation, which was in line with

**Figure 5. In vivo deletion of Dot1l affects specific myeloid and lymphoid precursor subsets.**
**(A)** Schematic overview of the experimental setup. Cre-ER$^{T2}$ *Dot1l*$^{wt/wt}$ and Cre-ER$^{T2}$ *Dot1l*$^{fl/fl}$ mice were injected with tamoxifen on three subsequent days to induce KO of *Dot1l*. On day 12, BM was harvested and analyzed. **(B)** Comparison of myeloid and lymphoid precursor percentages of Live Lineage$^{neg}$ AF$^{neg}$ cells in *Dot1l*-WT (Cre-ER$^{T2}$ *Dot1l*$^{wt/wt}$) and *Dot1l*-KO (Cre-ER$^{T2}$ *Dot1l*$^{fl/fl}$) conditions. CMPs, MDPs, and CDPs were gated from MHC class II$^{neg}$ CD11c$^{neg}$ CD135$^{pos}$ cells and defined as CD117$^{hi}$ CD115$^{neg}$ CMPs, CD117$^{int}$ CD115$^{hi}$ MDPs, and CD117$^{neg}$ CD115$^{hi}$ CDPs. CLPs were defined as MHC class II$^{neg}$ CD11c$^{neg}$ CD135$^{neg}$ CD127$^{pos}$. Pre-pDCs were gated as CD117$^{neg}$ CD135$^{pos}$ CD11c$^{pos}$ Siglec-H$^{pos}$ Ly-6D$^{pos}$, pDCs as B220$^{pos}$ Siglec-H$^{pos}$ BST2$^{pos}$, and cDC2s as B220$^{neg}$ MHC class II$^{hi}$ CD11c$^{hi}$ Sirpα$^{pos}$. **(C)** Comparisons of specific markers on pDCs and DC2s that were differentially expressed between *Dot1l*-WT and *Dot1l*-KO conditions based on the MFI of (N = 10) biological replicates. **(D)** On day 12, BM cells were stimulated with class A CpGs (ODN1585). IFNα levels were determined in the supernatant using ELISA. In addition, cells were stained intracellularly for IFNαβ. After overnight stimulation, cells were also stained for maturation markers. Conditions are shown with and without the addition of CpGs. The individual dots represent individual mice with a total of two independent experiments performed. **(E)** Pie charts highlighting the number of up- and down-regulated genes in *Dot1l*-KO DC populations compared with *Dot1l*-WT as determined by RNAseq (based on Log$_2$FC < −0.5 and >0.5 for the leftmost plot and <−1.0 and >1.0 for all others). **(F)** GO enrichment of genes that were down-regulated in KO pDCs (Log$_2$FC < −1.0). **(G)** Heatmap of selected transcription factors showing the H3K79me2 signal near the transcriptional start site (TSS; N = 2, green gradient) and expression differences between *Dot1l* and WT (N−1, blue-red gradient) in the sorted DC subsets. **(H)** GO enrichment of genes that were up-regulated in expression in all sorted subsets (Log$_2$FC > 0.5). For (F, H), pathways were ranked by fold enrichment and visualized using ShinyGO (121). **(I)** Heatmap correlating the H3K79me2 TSS signal to expression differences for genes from pathways enriched in all KO subsets (as visualized in (H)). Error bars indicate the mean ± SD. *P < 0.05, **P < 0.01, ***P < 0.001, ****P < 0.0001.

the decreased pDC frequencies. Ex vivo analysis of in vivo *Dot1l*-KO BM after 12 d demonstrated a decrease in myeloid precursors and an increase in cDC2s and lymphoid precursors, but unexpectedly no changes in the frequency of pDCs. However, analysis of sorted pDCs at this time point revealed that the cells had changed their phenotype in the absence of DOT1L. Pathways related to "*negative regulation of myeloid differentiation*" were specifically down-regulated based on GO enrichment in *Dot1l*-KO pDCs. In addition, the highly methylated transcription factors important for pDC differentiation, such as *Irf8*, *SpiB*, *Tcf4*, and *Bcl11a*, showed a decrease of expression in the *Dot1l*-KO pDCs. Because these transcription factors independently have been shown to be essential for pDC development and function, our results suggest that DOT1L-mediated methylation and transcription of these genes may be involved in the decrease of pDC differentiation observed in vitro (71, 72, 73, 74, 75, 76, 77).

Very recent studies describe a close relationship between preDC2A and pDCs and show that preDC2A expresses pDC transcription factors and markers such as Tcf4, BST2, and Siglec-H and also produces IFN upon class A CpG stimulation (93). Although we included B220 in the pDC gating in most of the experiments, including the detection of intracellular IFNαβ, and thereby selected for genuine pDCs and not preDC2A, this was not included in the sorting for the results shown in Figs 1 and 5. Therefore, we cannot exclude the possibility of contamination of preDC2A in the pDC gate in our ChIPseq and RNAseq experiments. Our observation that loss of DOT1L affects both pDC and DC2 differentiation is interesting considering their recently described shared features (93), but future studies with more elaborate flow cytometry panels and longer time points will be necessary to more precisely determine the role of DOT1L in the specific differentiation trajectories of the cDC2A and B subsets, as well as DC3 and other myeloid cell types.

The differential role of DOT1L in DC subsets is likely not linked to differences in *Dot1l* expression. Analysis of *Dot1l* expression in 86 primary cell types that span the mouse immune system (78) showed that the expression of *Dot1l* mRNA in DCs and their precursors clusters around the median expression level of all immune cell types (Fig S1A). Indeed, DOT1L seems to be constitutively active in many cell types and the global levels of H3K79 methylation deviate only modestly between cell types (98). Moreover, cell type–specific roles of DOT1L are generally not directly related to DOT1L expression levels. For example, DOT1L is a druggable dependency in MLL-rearranged leukemia and safeguards CD8 T cell differentiation, but these roles are not associated with a higher expression level of DOT1L (42, 61).

The role of DOT1L in DC cell subsets might involve a complex interplay between effects on differentiation and cell cycle progression. DOT1L has been reported to affect cell cycle progression, but this role is context-dependent and opposing roles have been described. Although some cells require DOT1L for cell cycle progression (e.g., germinal center B cells and several cancer cell types (40, 43, 61, 64, 99, 100, 101, 102, 103)), proliferation of some cell types is not affected by loss of DOT1L (e.g., CD8 T cells (49)), and other cells require DOT1L for cell cycle exit (e.g., early cardiomyocytes (104)). In several cases in which DOT1L loss leads to reduced proliferation, the cell cycle effect is linked to the induction of a

nonproliferative differentiation program, as we and others observed in germinal center B cells and GCB-derived diffuse large B cell lymphoma (40, 43, 63, 64) and as has been observed in MLL-rearranged leukemia treated with DOT1L inhibitors (61). If *Dot1l* KO would induce a DC subset–specific decrease in cell proliferation, this could, for example, cause a shift in subsets. However, we did not observe consistent changes in the mRNA expression of cell cycle genes (Table S3) and instead observed an increase in protein levels of S-phase progression marker Ki67 in MDP, CDP, and pre-pDC precursors in the BM (Fig S5E). Therefore, a decrease in cell proliferation is likely not the main cause of the loss of differentiated pDCs, but increased proliferation of precursors could be involved in the observed lack of pDC defects in the in vivo experiments.

Although DOT1L mainly acts as a histone methyltransferase with specificity toward H3K79me, other noncanonical functions have been described as well (65, 66, 67, 68, 86, 94, 105) and such functions may also be at play in DCs. To potentially differentiate between the two, we performed the in vitro expansion using the highly specific DOT1L inhibitor SGC-0946 (59). Supplementing our in vitro BM cultures with SGC-0946 generally recapitulated the results of our *Dot1l*-KO experiments and thereby confirmed that the effect of DOT1L on DC differentiation is related to its catalytic activity most likely via H3K79me. Because H3K79me is mainly lost via dilution because of proliferation, the effects of *Dot1l* KO can only be expected after a substantial number of cell divisions (49, 83, 106, 107, 108, 109, 110). A possible explanation for the discrepancy in pDC generation between in vitro BM cultures and in vivo experiments is the amount of proliferation that occurs; the in vivo time points of 3 and 12 d are relatively early considering the lifespan of DCs. This could be especially relevant for pDCs, which are proposed to have a longer half-life than cDCs in mice, and this potentially could be of importance for changes in cDC2 and pDC in vitro versus in vivo (111). The addition of SCF to FLT3L BM cultures stimulates proliferation of c-KIT⁺ progenitors, and this may drive faster loss of H3K79me in vitro than in vivo (80, 82). Finally, compensatory mechanisms specific for in vivo *Dot1l* KO might compensate for the loss of specific pDC precursors, as we observed higher Ki67 expression in *Dot1l*-KO pDC precursors indicating increased proliferation. Because of the harshness of the H3K79me2 staining, which requires treatment with SDS to expose the H3K79 epitope on the nucleosome surface (40, 42, 79), we could unfortunately not costain with the markers needed to identify specific DC subsets, but further experiments with sorted DC subsets and later time points will be necessary to demonstrate H3K79me loss and determine potential effects on pDCs in vivo.

Genes that were up-regulated in *Dot1l*-KO conditions often contained low or no detectable H3K79me2 and are therefore likely indirect targets of DOT1L (53, 54, 57). A clear up-regulation was observed in all DC subsets regarding genes involved in antigen presentation and immune response pathways. Multiple activation-related genes (*Cd40*, *Cd80*, *Cd86*), complement factors (*C1qa*, *C1qb*, *C1qc*), and MHC class II–related genes (*H2-ab1*, *H2-aa*, *H2-dmb2*) were up-regulated in all DC subsets. Indeed, increased surface-level expression of MHC class II was observed for pDCs and to a lower extent for cDC2s. CIITA is a known

regulator of MHC class II expression, and its expression is controlled by the negative regulator FBXO11 ([97], [112]). We observed H3K79me2 hypermethylation of *Fbxo11* in all DC subsets, whereas *Ciita* was relatively hypomethylated. Based on these results, we speculate that DOT1L might indirectly regulate MHC class II expression via this pathway. It should be noted that we cannot exclude cell extrinsic effects (e.g., cytokine production by other cell subsets) as the cause for increased antigen presentation and immune response pathways in the *Dot1l*-KO DCs. However, these results are in line with previous studies on inflammation regulation by DOT1L. A study on the function of DOT1L in atherosclerotic plaque macrophages, for example, reported that DOT1L deficiency or inhibition results in macrophage hyperactivation and MHC class II up-regulation, and linked DOT1L to the regulation of inflammatory processes in these macrophages ([45]). Tang et al proposed that intestinal immune tolerance is stimulated via increased DOT1L expression and H3K79me2 in DCs ([51]). In addition, in pancreatic and colon cancer mouse models, H3K79me2 controls FOXM1 expression, which is associated with the regulation of DC maturation ([50]), indicating a role of DOT1L in DC function in different models.

Here, we focus our epigenetics analyses on the correlation between H3K79me2 around and downstream of the TSS and how this relates to transcriptional changes in *Dot1l* KO. However, we cannot exclude that loss of DOT1L and subsequent loss of H3K79me affect the epigenetic cell state by other mechanisms such as changes in transcription elongation or broader chromatin effects in gene bodies. H3K79me2 is a co-transcriptional histone modification associated with active transcription, and the level of H3K79me2 typically correlates with the level of transcription. However, if and how H3K79me2 in turn affects gene promoters and transcription is still poorly understood. Although most observations suggest that H3K79me2 promotes transcription ([44], [49], [52], [58], [113], [114]), the role of DOT1L in transcription elongation is still under debate ([68], [115] Preprint), and several studies provide evidence for roles of DOT1L independent of its catalytic activity (see above). One major challenge in linking H3K79me to gene regulation is the slow turnover of the modification, which severely complicates identifying the direct targets of DOT1L that require H3K79me2 for proper gene regulation.

Previous studies investigating epigenetic regulators showed a dominant effect of HDAC1, but not HDAC2, on pDC and cDC2 development and of HDAC3 on pDC development ([35], [37]). A recent study indicated a role of DOT1L upstream of HDAC1 and positive regulation of DOT1L expression by HDAC1 ([116]). In contrast, we previously detected negative regulation of DOT1L activity by HDAC1 ([39], [95]). In our data, we observed considerable H3K79 methylation of the genes encoding HDAC1-3 in all DC subsets, leaving the possibility open that DOT1L might directly regulate the expression of HDACs in DCs. However, the effect of *Dot1l* KO on HDAC expression was variable between subsets. Further research would be required to properly investigate the interaction of DOT1L with HDACs in DCs and its potential effect on their differentiation.

Overall, our study reveals a differential role of DOT1L catalytic activity in the development of myeloid precursors, cDC2, and pDC in in vitro and in vivo settings. Furthermore, in vivo experiments showed that *Dot1l* deficiency results in an elevated inflammatory status of DCs. These findings form the basis for more mechanistic

studies investigating how DOT1L influences DC subset differentiation and function.

# Materials and Methods

### Mice

Cre-ER$^{T2}$ C57BL/6 mice contained Cre fused to the human estrogen receptor (Cre-ER$^{T2}$) integrated at the ROSA-26 locus, as described elsewhere ([117]). This strain was crossed with Dot1l$^{fl/fl}$ mice from the Dot1ltm1a(KOMP)Wtsi line as generated by the Wellcome Trust Sanger Institute (WTSI) and obtained from the KOMP Repository (www.komp.org) ([118]) and as described previously to generate the Cre-ERT2;*Dot1l*$^{fl/fl}$ strain ([39], [42]). Cre-ER$^{T2}$;*Dot1l*$^{fl/fl}$;OT-I mice were generated by crossing Cre-ER$^{T2}$;*Dot1l*$^{fl/fl}$ mice with OT-I (B6J carrying the OT-I T cell receptor transgenes) mice (kindly gifted by the Ton Schumacher group, originally from Jackson labs). Cre-ER$^{T2}$; *Dot1l*$^{wt/wt}$ (WT) and Cre-ER$^{T2}$;*Dot1l*$^{fl/fl}$ mice were used for experiments, without preselection based on OT-I presence. Experiments were performed in-house at the animal facility of the NKI. The mice were between 8 wk and 8 mo old and matched for age and gender. To induce the genetic ablation of DOT1L in vivo, Cre-ERT2;*Dot1l*$^{wt/wt}$ mice or Cre-ERT2;*Dot1l*$^{fl/fl}$ mice were injected intraperitoneally with 75 mg/kg tamoxifen (Sigma-Aldrich; an overview of key reagents used in this study can be found in Table S5) for three consecutive days. Experiments were performed according to institutional, national, and European guidelines as approved by the Animal Ethics Committee of the NKI (CCD number: AVD30100202215981).

### BM cultures

Murine BM cells (from C57BL/6 mice, Cre-ER$^{T2}$;*Dot1l*$^{wt/wt}$ mice, or Cre-ER$^{T2}$;*Dot1l*$^{fl/fl}$ mice; indicated in-text) were isolated from femurs and tibias by flushing the bones using a 10-ml syringe and 25G needle. The cells were filtered through a 70-µm cell strainer and concentrated by centrifugation at 323$g$ for 10 min with low break. Next, the cells were cultured at a concentration of 1 million cells per mL in 2 ml DC medium. DC medium consisted of RPMI-1640 (Gibco) supplemented with 10% heat-inactivated FCS (Biowest), 50 U/ml penicillin (Lonza), 50 µg/ml streptomycin (Lonza), 50 µM $\beta$-mercaptoethanol (Gibco), and 100× diluted GlutaMAX (Gibco), as well as 200 ng/ml FLT3L (PeproTech) and 50 ng/ml SCF (PeproTech) to expand FLT3-negative hematopoietic stem cells and DC progenitors ([82]). On day 3 after the start of the cultures, each well was divided over two wells and 1 ml fresh DC medium without SCF was added. On day 7, the cells were harvested, and each well was washed with 1 ml PBS two times. After centrifugation for 5 min at 505$g$, the cells were stained using a flow cytometry panel and acquired on the Aurora spectral flow cytometer (Cytek). Where indicated, tamoxifen or DOT1L inhibitor was added to the cultures. 4-Hydroxytamoxifen (Sigma-Aldrich) was added on day 0 at a concentration of 10 nM. DOT1L inhibitor SGC-0946 (Selleckchem) was added on days 0, 3, and 5 at a concentration of 1 µM. The same volume of DMSO was added to the control wells.

 Life Science Alliance

## H3K79me2 staining and Dot1l PCR

BM suspensions were prepared as described above. Splenocytes were isolated by meshing spleens three times through a 70-$\mu$m strainer (BD Falcon) to create single-cell suspensions. The samples were incubated with lysis buffer (150 mM $NH_4Cl$, 10 mM $KHCO_3$, 0.2 mM EDTA) for 3 min on ice to lyse erythrocytes. For flow cytometry, $2.5 \times 10^5$ cells per sample were washed with PBS and subsequently stained with Zombie NIR (BioLegend). Intracellular staining was performed using the Transcription Factor Buffer kit (BD Biosciences), according to the manufacturer's protocol. Cells were stained with anti-H3K79me2 (Millipore) in 0.25% SDS-supplemented perm/wash buffer (BD Biosciences) followed by 1:1,000 secondary antibody staining (goat anti-rabbit AF488; Invitrogen) as described previously (42). Samples were measured using the LSRFortessa II (BD Biosciences) and analyzed using FlowJo V10.9.0 (BD Biosciences).

To confirm the deletion of exon 2 of *Dot1l* by PCR, genomic DNA was isolated using ISOLATE II Genomic DNA Kit (Bioline) as per the manufacturer's protocol. *Dot1l*$^{WT}$, *Dot1l*$^{fl}$, and *Dot1l*$^{\Delta}$ (exon 2) alleles were detected as described previously (42), using the following primers: Dot1l_FWD: GCAAGCCTACAGCCTTCATC, Dot1l_REV: CACCGG ATAGTCTCAATAATCTCA, and Dot1l_Δ: GAACCACAGGATGCTTCAG. PCRs were performed using MyTaq Red Mix (GC Biotech). Agarose gel electrophoresis was performed to determine the genotype. The efficacy of floxing out exon 2 was determined by quantifying band densities using ImageJ (version 1.54g), correcting for product size and dividing the band density of the *Dot1l*$^{\Delta}$ band over the total *Dot1l* band density within the sample.

## Sample preparation for in vivo experiments

Bone marrow and spleen suspensions were prepared from Cre-ER$^{T2}$; *Dot1l*$^{wt/wt}$ mice or Cre-ER$^{T2}$;*Dot1l*$^{fl/fl}$ mice. Spleens were mechanistically dissociated and digested with 4 mg/ml lidocaine (Sigma-Aldrich), 4 Wunsch units/ml Liberase TL (Roche), and 50 mg/ml DNase I (Roche) for 20 min at 37°C with continuous stirring. Digestion was then halted by the addition of cold medium (RPMI-1640 [Gibco] with 10% heat-inactivated FCS [Biowest], 10 mM ethylenediaminetetraacetic acid [EDTA], 20 mM Hepes, and 50 $\mu$M 2-mercaptoethanol) and another 10 min of stirring at 4°C. Ammonium–chloride–potassium lysis buffer was added to lyse the red blood cells, and cells were filtered through a 70- to 100-$\mu$m cell strainer immediately afterward. The cells were then counted, and a flow cytometry staining and functional experiments were performed.

## Flow cytometry panels

To identify DC subsets in the BM as described in Fig 1, we first incubated the cells with 10 $\mu$g/ml of anti-CD16/32 (clone 2.4G2, in-house production) and Fixable Viability Dye eFluor 780 (eBioscience). Next, we stained the cells for 20 min at 4°C with the following combination of antibodies: anti-XCR1 (clone ZET; BioLegend), anti-MHC class II (clone M5/114.15.2; BioLegend), anti-CD11c (clone N418; BioLegend), anti-BST2 (clone 927; BioLegend), anti-CD3 (clone 17A2; BioLegend), anti-CD19 (clone 6D5; BioLegend), anti-NK1.1 (clone PK136; BioLegend), anti-Siglec-H (clone REA819; Miltenyi Biotec),

and anti-Sirp$\alpha$ (clone P84; BioLegend). Finally, the cells were fixed using 2% PFA (Electron Microscopy Sciences) and subsequently acquired on the Aurora spectral flow cytometer (Cytek).

To identify precursor cell subsets in the BM as described in Figs 2, 3, and 4, single-cell suspensions were incubated with 10 $\mu$g/ml of anti-CD16/32 (clone 2.4G2, in-house production) and LIVE/DEAD Fixable Blue Dead Cell Stain (Thermo Fisher Scientific). Afterward, the cells were stained with the following antibody panel for 20 min at 4°C: anti-MHC class II (clone M5/114.15.2; BioLegend), anti-CD117 (clone 2B8; BioLegend), anti-CD11c (clone N418; BioLegend), anti-BST2 (clone 927; BioLegend), anti-CD3 (clone 145-2C11; Immuno-Tools), anti-NK1.1 (clone PK136; ImmunoTools), anti-erythroid cells (clone TER-119; ImmunoTools), anti-CD90 (clone MRC OX-7; ImmunoTools), anti-Ly6G (clone RB6-8C5; ImmunoTools), anti-B220 (clone RA3-6B2; BioLegend), anti-CD135 (clone A2F10; Invitrogen), anti-CD127 (clone A7R34; BioLegend), anti-CD19 (clone 6D5; BioLegend), anti-CD115 (clone AFS98; BioLegend), anti-Sirp$\alpha$ (clone P84; BioLegend), anti-Siglec-H (clone REA819; Miltenyi Biotec), and anti-Ly6D (clone REA906; Miltenyi Biotec). The cells were subsequently fixed using 2% PFA (Electron Microscopy Sciences) and acquired on the Aurora spectral flow cytometer (Cytek).

To identify immune cell subsets in the spleen as described in Fig 5, splenocytes were incubated with 10 $\mu$g/ml of anti-CD16/32 (clone 2.4G2, in-house production) and Fixable Viability Dye eFluor 780 (eBioscience). Next, we stained the cells for 20 min at 4°C with the following combination of antibodies: anti-XCR1 (clone ZET; BioLegend), anti-Ly6C (clone HK1.4; BioLegend), anti-MCHII (clone M5/114.15.2; BioLegend), anti-CX3CR1 (clone SA011F11; BioLegend), anti-CD11c (clone N418; BioLegend), anti-BST2 (clone 927; BioLegend), anti-CD11b (clone M1/70; BioLegend), anti-CD3 (clone 145-2C11; ImmunoTools), anti-NK1.1 (clone PK136; ImmunoTools), anti-B220 (clone RA3-6B2; BioLegend), anti-Ly6G (clone 1A8; BioLegend), anti-F4/80 (clone BM8; BioLegend), anti-CD19 (clone 6D5; BioLegend), anti-Siglec-H (clone REA819; Miltenyi Biotec), and anti-Sirp$\alpha$ (clone P84; BioLegend). Finally, the cells were fixed using 2% PFA (Electron Microscopy Sciences) and subsequently acquired on the Aurora spectral flow cytometer (Cytek).

## Class A CpG stimulation of single-cell suspensions

Where indicated, BM cells and/or splenocytes were stimulated with class A CpGs (ODN1585; InvivoGen). $3 \times 10^6$ BM cells or splenocytes were seeded in a round-bottom 96-well plate with medium (RPMI-1640 [Gibco] with 10% heat-inactivated FCS [Biowest], 50 U/ml penicillin [Lonza], 50 mg/ml streptomycin [Lonza], and 50 $\mu$M 2-mercaptoethanol) containing 1 $\mu$M ODN1585 (InvivoGen). After overnight incubation, the supernatant was harvested for ELISA, and the cells were stained with the previously described panels plus anti-CD86 (clone GL1; BD Biosciences).

To analyze the IFN$\alpha$ production in the supernatant of these cultures, MaxiSorp ELISA plates (NUNC) were coated in coating buffer (pH 9.2) with 200x diluted anti-IFN$\alpha$1 capture antibody (BioLegend) overnight. After, the plates were washed three times with PBS containing 0.05% Tween-20. Next, the wells were blocked with 1% PBS-BSA for 30 min at 37°C. 100× (BM samples) or 5× (spleen samples) diluted supernatant or IFN$\alpha$1 standard

(BioLegend) was added to the appropriate wells and incubated for 2 h on a shaking plate at RT. The plates were washed four times with PBS–Tween-20 and subsequently incubated with 200× diluted anti-IFNα1 detection antibody (BioLegend) for 1 h at RT. The plates were washed four times with PBS–Tween-20 and incubated with 1,000× diluted Avidin-HRP (BioLegend). After washing again for five times with PBS–Tween-20, TMB (100 µg/ml) was added as a substrate and absorbance was measured at 450 nm with a microplate absorbance spectrophotometer (Bio-Rad).

To determine the intracellular production of IFNαβ, $3 × 10^6$ BM cells or splenocytes were seeded in a round-bottom 96-well plate with medium (RPMI-1640 [Gibco, Life Technologies] with 10% heat-inactivated FCS [Biowest], 50 U/ml penicillin [Lonza], 50 mg/ml streptomycin [Lonza], and 50 µM 2-mercaptoethanol) containing 1 µM of ODN1585 (InvivoGen). After 3-h incubation, GolgiPlug (BD Biosciences) was added for an additional 2 h. The cells were then stained extracellularly with the panels as described above and fixed using 2% PFA (Electron Microscopy Sciences). After two washes with 0.5% saponin, an intracellular staining was performed with a combination of anti-IFNα (clone RMMA-1; R&D Systems; labeled in-house with AF647) and anti-IFNβ (clone RMMB-1; R&D Systems; labeled in-house with AF647) in 0.5% saponin for 30 min at 4°C.

### Cell sorting, RNA-sequencing preparation, and data processing

*Dot1l* was genetically ablated in vivo via tamoxifen injections as described above. On day 12 after the first tamoxifen injection, BM cells were isolated as described above and pooled per group to prepare for sorting (*Dot1l* WT versus *Dot1l* KO). The cells were resuspended in MACS buffer (PBS, 0.5% BSA, and 2 mM EDTA) and incubated with a mixture of CD11c and BST2 MicroBeads UltraPure ($100 µl$ per $10^8$ cells; Miltenyi Biotec) for 15 min at 4°C for positive selection of CD11c+ and BST2+ cells. The cells were then washed with MACS buffer and loaded onto a MACS LS column (Miltenyi Biotec) in a MACS Separator according to the instructions of the manufacturer. The selected cells were incubated with 10 µg/ml of anti-CD16/32 (clone 2.4G2, in-house production) and Fixable Viability Dye eFluor 780 (eBioscience). Next, we stained the cells for 20 min at 4°C with the following combination of antibodies: anti-XCR1 (clone ZET; BioLegend), anti-CD11c (clone N418; BioLegend), anti-BST2 (clone 927; BioLegend), anti-Siglec-H (clone 440c; BD Biosciences), and anti-Sirpα (clone P84; BD Biosciences). Immediately after this, cDC1s, cDC2s, and pDCs were sorted from the *Dot1l*-WT and the *Dot1l*-KO group using BD FACSAria Fusion Flow Cytometer (BD Biosciences). pDCs were sorted as CD11c+ BST2+ Siglec-H+, cDC1s were sorted as CD11c+ BST2− Siglec-H− XCR1+, and cDC2s were sorted as CD11c+ BST2− Siglec-H− Sirpα+.

Sorted cells were pelleted at 1,000 RCF for 5 min, washed once with 1 ml ice-cold PBS, transferred to RNase/DNase-free Eppendorf tubes, and pelleted once more (1,000 RCF, 4°C, 10 min). Pellets were resuspended in RLT buffer (QIAGEN) supplemented 1:100 with 2-mercaptoethanol (14.3 M; Sigma-Aldrich) and stored at −80°C. Total RNA was isolated using the RNeasy Mini kit (QIAGEN) according to the manufacturer's protocol. Quality control was performed on the 2,100 Bioanalyzer using the RNA NanoChip (Agilent), and samples were library-amplified using SMART-Seq

v4 Ultra Low Input RNA Kit (Takara) and quality-checked on the 2,100 Bioanalyzer using the DNA 7500 chip (Agilent). Samples were sequenced (51 bp, paired end) using the NovaSeq 6000. Reads were aligned against the mouse reference genome (GRCm38/mm10) using TopHat (version 2.1, bowtie version 1.1), supplied with a Gene Transfer Format (GTF, Ensembl version 77) file using the following parameters: "prefilter-multihits –no-coverage-search – bowtie1 –library-type fr-firststrand." A custom script based on HTSeq-count was used to count the number of uniquely mapped reads per gene, as listed in the provided Gene Transfer Format (GTF) file.

Subsequent analysis was performed in R version 4.2.2 with Bioconductor packages from release 3.16. The analysis was restricted to the genes that had at least 10 counts 3 or more samples to exclude very low-abundance genes. On these genes, a PCA was performed using the "prcomp" function on variance-stabilizing transformed data from the "vst" function of the DESeq2 package (version 1.38.3). Data were visualized using ggplot2 (version 3.5.1). GO enrichment analyses were performed and visualized using ShinyGO v0.82, hosted by the South Dakota State University, by filtering for a minimum pathway size of 20 and ranking based on fold enrichment but otherwise using default settings (119, 120, 121). For these analyses, only genes that had an expression of at least $2^5$ counts per million reads in either WT or KO, as well as a $Log_2FC$ of >1 or < −1 (unless otherwise indicated), were considered. The data obtained from these analyses can be found in Table S4. To facilitate quantitative comparisons with the H3K79me2 ChIPseq data (see below), the mean library size–corrected expression value for each gene between all samples (BaseMean) was calculated and where relevant (for Fig 1G) further normalized for transcript length using the "fpkm" function of DESeq2 (version 1.38.3).

### Normalized H3K79me2 ChIPseq preparation and data processing

BM cells from WT C57BL/6J mice were sorted as described above into pDC, cDC1, and cDC2 pools, three pools each with each pool consisting of cells from five mice. Sorted cells were cross-linked for 10 min at room temperature in RPMI supplemented with 1% methanol-free formaldehyde (Thermo Fisher Scientific) and subsequently quenched by adding glycine (final concentration 125 mM) for 5 min. Samples were then washed twice using PBS + protease inhibitor cocktail (PIC; Roche) + 10% FBS (Capricorn), aspirated, and resuspended in ice-cold nucleus lysis buffer (50 mM Tris–HCl, pH 8, 10 mM EDTA, pH 8, 1% SDS + PIC) for 10 min. Samples were sonicated for 3 min (3 × 30 s on, 30 s off) using the Bioruptor (Pico), after which debris was pelleted (4°C, 13,000 RCF, 10 min) and supernatant was transferred to a new Eppendorf tube. Samples were supplemented with sample buffer (nine parts ChIP dilution buffer [50 mM Tris–HCl, pH 8, 0.167 M NaCl, 1.1% Triton X-100, 0.11% sodium deoxycholate] and 5 parts RIPA-150 [50 mM Tris–HCl, pH 8, 0.15 M NaCl, 1 mM EDTA, pH 8, 0.1% SDS, 1% Triton X-100, 0.1% sodium deoxycholate + PIC] per 1 part sample). Ten percentage of each suspension was de-cross-linked by adding 10 mg/ml RNase A (Sigma-Aldrich) and 10 mg/ml Proteinase K (Sigma-Aldrich) and incubating for 1 h at 50°C, then overnight at 65°C. From these de-cross-linked samples, the shearing efficacy was checked by gel

electrophoresis, and gDNA was isolated and quantified (as described above).

A yeast chromatin spike-in was added 1:1,000 to all samples based on DNA concentration. This spike-in was generated using a *dot1Δ*, H3-3xHA (HHT2-3xHA) yeast strain prepared as described previously (49). Samples were homogenized in sample buffer, and 10% was stored as input, whereas 90% was used for simultaneous IP for HA (yeast spike-in) and H3K79me2, using 0.5 $\mu$g mouse anti-HA antibody (clone 12CA5, Protein Facility NKI) and 2 $\mu$l rabbit anti-H3K79me2 (Millipore) per sample. Samples were incubated overnight under rotation at 4°C, after which Dynabeads Protein G (Thermo Fisher Scientific) in sample buffer was added for 2 h. The supernatant was aspirated using the DynaMag-2 magnetic rack (Thermo Fisher Scientific), and samples were washed once with RIPA-150 (50 mM Tris–HCl, pH 8, 0.15 M NaCl, 1 mM EDTA, pH 8, 0.1% SDS, 1% Triton X-100, 0.1% sodium deoxycholate), twice with RIPA-500 (50 mM Tris–HCl, pH 8, 0.5 M NaCl, 1 mM EDTA, pH 8, 0.1% SDS, 1% Triton X-100, 0.1% sodium deoxycholate), twice with RIPA-LiCl (50 mM Tris–HCl, pH 8, 1 mM EDTA, pH 8, 1% Nonidet P-40, 0.7% sodium deoxycholate, 0.5 M LiCl$_2$), and once with TE. Samples were resuspended in 150 $\mu$l direct elution buffer (10 mM Tris–HCl, pH 8, 0.3 M NaCl, 5 mM EDTA, pH 8, 0.5% SDS), de-cross-linked, and quantified as described above. Samples were library-amplified, pooled, and sequenced (51 bp, paired end) on the NovaSeq 6000. Reads were aligned to a concatenated mouse (GRCm38/mm10) and yeast (sacCer3) genome, on which peaks were called using MACS2 version 2.2.9.1 with the arguments "-f BEDPE –g mm –keep-dup all --broad." Data were further normalized using the spike-in reads and visualized as described previously (49).

PCA was performed on the ChIPseq signal near the transcriptional start site (TSS; + 2 kb) using the "prcomp" function as described above. Peaks that uniquely mapped to a single-sorted DC subtype were further analyzed with the PubMed enrichment built into the STRING database using default arguments (122). The output of the STRING queries can be found in Table S1. Heatmaps and correlation plots with H3K79me2 signal near the TSS were visualized using GraphPad Prism (version 10.0.2).

### Analysis of flow cytometry data

All data were unmixed after acquisition using SpectroFlo software (Cytek). Further analysis was performed using OMIQ by Dotmatics. Manual gating was used to identify cell subsets after removing debris and gating on live cells. Data were generated on the geometric mean fluorescence intensity (gMFI) of specific markers or percentage change of target populations, and this was plotted in GraphPad Prism v8 (GraphPad).

### Statistical analysis

To identify statistical differences between two groups, a two-tailed *t* test was used. For the DOT1L inhibitor experiments, a paired two-tailed *t* test was used. For three or more groups, a one-way or two-way ANOVA with a Šidák post hoc test was performed. All statistical analyses were performed in GraphPad Prism v8 (GraphPad). For all enrichment analyses (ShinyGO, GO enrichments, and STRING analysis), multiple comparison corrections were performed using the platform default settings.

## Data Availability

All processed data are in the main text or the supplementary materials. RNAseq and ChIPseq data have been deposited in the Gene Expression Omnibus (GEO) and are publicly available via accessions GSE301937 and GSE301939.

## Supplementary Information

## Acknowledgements

This work was supported by NWO ZonMW (TOP91218024 to JMM den Haan; TOP91218022 to F van Leeuwen), the Dutch Cancer Society (2019-12802, 2021-2/14093, 2022-14453 to JMM den Haan; NKI2018-1/11490, 2022-2 EXPL/14479, 17005/2025-EXPL to F van Leeuwen), from Health Holland TKI-PPP to JMM den Haan, and institutional grants to the Netherlands Cancer Institute from the Dutch Cancer Society (KWF) and the Dutch Ministry of Health, Welfare and Sports. AJ Affandi and JGC Stolwijk were supported by NWO Veni ZonMw (09150162010163) and Stichting Cancer Center Amsterdam (CCA 2022-9-83). AZ Wang was supported by the Chinese Scholarship Council (202006230076). Some figure panels were created in BioRender under the license of F van Leeuwen (2026) and are accessible via https://BioRender.com/ldsy7wx. We would like to thank the staff of the Netherlands Cancer Institute Preclinical Intervention Unit of the Mouse Clinic for Cancer and Ageing (MCCA) for the animal care and technical support. We acknowledge the Microscopy and Cytometry Core Facility (MCCF) at the Amsterdam UMC (Location VUmc) and the Flow Cytometry Facility of the NKI for providing assistance and advice with flow cytometry experiments. We would like to thank the Genome Core Facility (GCF) of the NKI for technical support for the RNAseq and ChIPseq experiments, and Teun van de Brand for support with subsequent analysis. We thank Heinz Jacobs for valuable discussions and input during the studies.

### Author Contributions

RG Bouma: conceptualization, data curation, formal analysis, validation, investigation, visualization, methodology, and writing—original draft, review, and editing.

W-J de Leeuw: conceptualization, data curation, software, formal analysis, validation, investigation, visualization, methodology, and writing—original draft, review, and editing.

AZ Wang: investigation.

M Malik: conceptualization, formal analysis, and validation.

JGC Stolwijk: investigation.

VAL Konijn: investigation.

A Mensink: data curation, software, formal analysis, and visualization.

N Proost: methodology.

MK Nijen Twilhaar: investigation.

T van Welsem: investigation and methodology.

N Seyed Toutounchi: investigation.

AJ Affandi: investigation and methodology.

JT van Dinter: data curation, software, and formal analysis.

F van Leeuwen: conceptualization, supervision, funding acquisition, and writing—original draft, review, and editing.

JMM den Haan: conceptualization, supervision, funding acquisition, and writing—original draft, review, and editing.

## Conflict of Interest Statement

The authors declare that they have no conflict of interest.

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
