## [Reviewer comments · Life Science Alliance]

Histone methyltransferase DOT1L differentially affects the development of dendritic cell subsets

Rianne Bouma, Willem-Jan de Leeuw, Aru Wang, Muddassir Malik, Joeke Stolwijk, Veronique Konijn, Anne Mensink, Natalie Proost, Maarten Nijen Twilhaar, Tibor van Welsem, Negisa Seyed Toutounchi, Alsya Affandi, Jip van Dinter, Fred van Leeuwen, and Joke den Haan

DOI: <https://doi.org/10.26508/lsa.202603624>

Corresponding author(s): Fred van Leeuwen, The Netherlands Cancer Institute and Joke den Haan, Amsterdam UMC Location Vrije Universiteit Amsterdam

Review Timeline:

Submission Date:	2026-01-09
Editorial Decision:	2026-02-18
Revision Received:	2026-02-25
Accepted:	2026-03-03

Scientific Editor: Tim Fessenden

Transaction Report:

Please note that the manuscript was reviewed at *Review Commons* and these reports were taken into account in the decision-making process at *Life Science Alliance*.

Review
COMMONS

Reviews

Review #1

1. Evidence, reproducibility and clarity:

****Summary:****

In this study, Bouma et al. investigate the epigenetic mechanisms involved in dendritic cell (DC) development, focusing on the role of the lysine methyltransferase DOT1L, which mediates histone H3 lysine 79 (H3K79) methylation. The authors first show that Dot1l is expressed across most DC subsets and their progenitors. Consistently, DOT1L activity was detected in these subsets, as ChIP-seq analysis revealed an enrichment of H3K79 methylation marks around the transcription start sites of numerous genes that regulate DC fate. These marks were associated with active transcription, as confirmed by RNA sequencing. To assess the functional role of Dot1l in DC development, the authors used Rosa26Cre-ERT2 × Dot1l^{flox/flox} mice. Bone marrow (BM) cells from these mice were treated in vitro with tamoxifen and cultured with FLT3L and SCF to induce DC differentiation. Dot1l deletion impaired the development of plasmacytoid DCs (pDCs) and enhanced the generation of conventional DC2 (cDC2), while leaving cDC1 development unaffected. Similarly, in vivo tamoxifen treatment of Rosa26Cre-ERT2 × Dot1l^{flox/flox} mice for three days led to a comparable impairment of DC development upon in vitro culture of BM cells. Beyond mature DCs, Dot1l deletion also disrupted the ability of BM cells to generate common myeloid progenitors (CMPs), monocyte-dendritic cell progenitors (MDPs), and common DC progenitors (CDPs). These effects were attributed to the methyltransferase activity of DOT1L, as pharmacological inhibition of DOT1L produced similar outcomes. Interestingly, while in vivo tamoxifen treatment altered the frequencies of progenitor populations (MDP, CDP, CMP) in the BM, it did not significantly change the frequency of pDCs in the BM or spleen. Moreover, an increase in the cDC2 population was observed only in the BM, with no effect detected in the spleen. With these findings the authors claim that epigenetic regulation of gene expression by DOT1L is important for proper dendritic cell development.

****Major comments:****

While this study demonstrates that DOT1L regulates DC development in vitro, its inducible deletion in vivo using tamoxifen does not appear to significantly affect the overall distribution or function of DCs. Therefore, further investigation is needed to clarify the role of DOT1L in regulating DC fate under physiological conditions. The authors analyzed DC populations at only two time points (3 and 12 days) following tamoxifen-induced Dot1l deletion. As noted in the discussion, these time points are relatively early considering the lifespan of DCs, which often extends beyond this period. It would thus be important to assess the effects of Dot1l deletion over a longer duration (e.g., at least one month) to fully evaluate its impact on DC development. In addition to the BM, an extensive analysis of DCs population should be carried in the spleen as well as lymph nodes.

Given the broad activity of the Rosa26-Cre system, prolonged deletion may affect overall mouse health and/or the function of other cell types that contribute to DC development; therefore, using a DC-specific Cre driver (e.g., CD11c-Cre) would provide a more targeted approach. Alternatively, competitive BM chimera experiments could be performed by reconstituting irradiated control mice with a 1:1 mixture of BM cells from Rosa26Cre-ERT2 × Dot1l^{flox/flox} and Rosa26Cre-ERT2 × Dot1l^{wt/flox} mice, both pre-treated with tamoxifen in vitro. Such experiments would offer more definitive evidence for the role of DOT1L in DC development in vivo.

Aside from this point, the data and methods are clearly presented, and the figures are largely self-explanatory. All experiments were adequately replicated three times. Statistical analyses were primarily performed using t-tests, and ANOVA with multiple comparisons when appropriate. Since these are parametric tests that assume a normal distribution, it would be important to confirm whether the analyzed samples meet this assumption. If not, non-parametric tests should be used instead.

****Minor comments:****

It would be informative to show how specific Dot1l expression is in DCs and their progenitors compared with other immune lineages (e.g., lymphocytes) and their precursors. The data suggest that DOT1L regulates H3K79 methylation of both shared and subset-specific genes among DC populations. The authors could elaborate on how this regulation achieves cell-type specificity—perhaps through differential Dot1l expression levels across DC subsets.

Interestingly, Dot1l deletion both in vitro and in vivo markedly reduces the frequency of common DC progenitors (CDPs), which give rise to cDC1 and cDC2. The authors should discuss how such a substantial loss of progenitors does not proportionally affect downstream cDC populations.

Although in vivo tamoxifen-induced deletion of Dot1l in Rosa26Cre-ERT2 × Dot1l^{flox/flox} mice does not significantly alter the overall distribution of DC subsets (pDCs and cDCs), it appears to modify their phenotype. It would therefore be valuable to examine how Dot1l loss impacts the functional properties of individual DC subsets. While pDC responsiveness to CpG stimulation seems preserved in the absence of Dot1l, assessing how cDCs respond to TLR3 and TLR4 stimulation and their capacity to activate T cells would provide important additional insights.

2. Significance:

Significance (Required)

General assessment: Bouma et al. present compelling evidence that DOT1L is an important regulator of DC differentiation in vitro from bone marrow-derived cells. They further demonstrate that DOT1L regulates DC development through its lysine methyltransferase activity, mediating histone H3K79 methylation. While these in vitro findings are robust and well supported, the physiological relevance of DOT1L function in vivo remains less clearly established. Additional experiments would help to strengthen the conclusions regarding its role under physiological conditions.

Advance: While numerous transcription factors have been described as key regulators of DC subset development and fate, the role of epigenetic regulation in this process remains relatively understudied and poorly understood. This study addresses this important gap in the literature and provides novel insights into the role of H3K79 methylation mediated by DOT1L in controlling DC development.

Audience: This paper will be of interest for a specialized audience in the field of the regulation of dendritic cell ontogeny. This work could influence additional research to investigate the epigenetic regulation of DCs development.

3. How much time do you estimate the authors will need to complete the suggested revisions:

Estimated time to Complete Revisions (Required)

(Decision Recommendation)

Between 3 and 6 months

4. Review Commons values the work of reviewers and encourages them to get credit for their work. Select 'Yes' below to register your reviewing activity at Web of Science Reviewer Recognition Service (formerly Publons); note that the content of your review will not be visible on Web of Science.

Yes

Review #2

1. Evidence, reproducibility and clarity:

Bouma et al. present a comprehensive analysis of DOT1L-mediated histone H3K79 methylation across canonical DC subsets. By mapping the methylation landscape, the authors demonstrate that DOT1L regulates both shared and subset-specific gene programs. They show that in vitro or in vivo deletion of Dot1l, followed by in vitro differentiation, results in reduced myeloid progenitors and pDCs alongside an increase in cDC2s, while cDC1 numbers remain largely unaffected. Functionally, Dot1l-deficient DCs fail to produce IFN α upon stimulation. Transcriptomic profiling reveals enrichment of antigen presentation pathways in Dot1l-KO subsets, with upregulated MHC class II surface expression in pDCs. Mechanistically, pharmacological inhibition of DOT1L links these effects to its methyltransferase activity. Collectively, the data suggest that DOT1L differentially regulates canonical DC subset development and represses antigen presentation pathways.

The manuscript is well-written and technically sound. However, several conclusions would benefit from deeper discussion or additional experimental validation.

****Major Comments****

1. Interpretation of DC balance changes and cell-cycle effects

The authors propose that DOT1L loss skews DC differentiation toward a pDC-like phenotype. However, DOT1L deletion or inhibition, and the consequent global loss of H3K79 methylation, is well known to downregulate key cell-cycle genes (e.g., Cyclin D1, Cyclin E, CDK4/6, MCM family) while upregulating cell-cycle inhibitors (e.g., Cdkn1a and b). These transcriptional changes are associated with slower proliferation, G1 arrest or delayed S-phase entry, and reduced DNA replication fork progression. Importantly, blocking DNA synthesis (e.g., with aphidicolin or mitomycin C) during early culture inhibits DC emergence, underscoring that proliferation is essential for differentiation.

The authors should discuss how their findings align with this established literature. Could the observed DC subset shifts result from impaired cell-cycle progression rather than lineage-specific transcriptional reprogramming? A more detailed consideration of this point is needed.

2. Discrepancy between in vitro and in vivo pDC phenotypes

The in vitro data show a marked reduction in pDCs, yet in vivo pDC numbers appear unchanged. Although the discussion briefly mentions proliferation differences, this discrepancy deserves a clearer explanation or experimental follow-up.

Minor Comments

- Clarify statistical methods, specify biological replicate numbers, and indicate whether corrections for multiple comparisons were applied to transcriptomic analyses.
- The introduction is somewhat lengthy and repetitive; condensing it would improve focus.
- In the discussion sometimes it is not clear the distinction between findings and speculation.
- Ensure consistent gene name formatting throughout (e.g., Dot1l, Dot1L).

2. Significance:

Significance (Required)

The current manuscript fills a gap in knowledge, and this is its major strength.

Other strengths are clarity and technical appropriateness.

The major weakness is that the work is mainly descriptive. Mechanistic insights into DOT1L-dependent transcriptional regulation are still weak.

The proposed mechanism -that DOT1L maintains pDC identity through H3K79 methylation at key transcription factors (Tcf4, SpiB, Irf8)- is intriguing but currently lacks functional evidence. The authors should consider validating this model experimentally, by modulating the expression of these genes without affecting DOT1L activity.

Also the model suggesting that DOT1L indirectly represses antigen presentation via the Fbxo11-Ciita pathway is interesting but remains speculative. Additional mechanistic data would help support this claim.

3. How much time do you estimate the authors will need to complete the suggested revisions:

Estimated time to Complete Revisions (Required)

(Decision Recommendation)

Between 1 and 3 months

4. Review Commons values the work of reviewers and encourages them to get credit for their work. Select 'Yes' below to register your reviewing activity at Web of Science Reviewer Recognition Service (formerly Publons); note that the content of your review will not be visible on Web of Science.

Yes

Review #3

1. Evidence, reproducibility and clarity:

This manuscript investigates the role of DOT1L and its H3K79 methyltransferase activity in dendritic cell (DC) differentiation. The authors employ a combination of in vitro FLT3L/SCF bone marrow culture systems, in vivo inducible knockout models, and genome-wide H3K79me2 ChIP-seq and RNA-seq analyses to demonstrate that DOT1L influences the balance between pDC and cDC2 differentiation, while leaving cDC1 development largely unaffected. The study further identifies transcriptional and epigenetic programs associated with these changes, linking DOT1L deficiency to altered antigen presentation pathways and loss of pDC-associated transcription factors.

The paper provides valuable insights into DC biology. However, some of the key conclusions rely heavily on in vitro systems and short-term tamoxifen deletion models, which limit the interpretation of the in vivo data. Strengthening or clearly defining these limitations would substantially improve the paper's impact and clarity.

Major Comments

1. To strengthen the paper, the authors could follow one of two alternative strategies:

(1) Validate their in vitro observations through in vivo experiments, or

(2) Focus on deepening and refining their in vitro findings, moving the limited in vivo data to the supplementary material and explicitly acknowledging the limitations of the tamoxifen-inducible system.

Strategy 1 - Strengthen in vivo validation

- The experiments presented in Figures 3 and 5 could be repeated in a competitive bone marrow chimera setting (e.g. CD45.1/CD45.2 irradiated hosts reconstituted with a 1:1 mix of WT CD45.1⁺ and Dot1l-KO CD45.2⁺ cells).

- This design would allow dissection of direct (cell-intrinsic) versus indirect effects of DOT1L deficiency and could mitigate confounding effects of incomplete or asynchronous deletion.

- After reconstitution, mice could be maintained on tamoxifen-supplemented chow for a longer period to ensure efficient recombination and adequate time for observing phenotypic consequences.

- Flow cytometric analysis of spleen and bone marrow should use more refined panels to explore DC precursor and subset deficiencies. Suggested reference panels: Rodrigues et al., *Immunity* 2024; Minutti et al., *Nat. Immunol.* 2024; Zhu et al., *Nat. Immunol.* 2015.

Strategy 2 - Refine in vitro system and reposition in vivo data

- The authors could replicate their differentiation assays under conditions that emulate the chimera approach by co-culturing WT (CD45.1⁺) and Dot1l-KO (CD45.2⁺) bone marrow cells.

- This would reveal potential competition or cross-talk between WT and mutant cells and provide clearer mechanistic insight into cell-intrinsic versus extrinsic effects.

- The authors should examine how tamoxifen itself affects differentiation and measure the kinetics of deletion and H3K79me loss to better contextualize the dynamic response.

- It would also be valuable to assess which cDC2 subtypes (A vs. B) are preferentially affected by Dot1l deficiency, again using more sophisticated flow cytometry panels (see references above).

If this in vitro-focused strategy is adopted, the in vivo data could be moved to the supplementary material, with explicit acknowledgment that the inducible deletion model and the gradual nature of H3K79me dilution limit the interpretation of the in vivo findings.

2. In Figures 2 and 3, the efficiency of H3K79me2 depletion following Dot1l excision should be assessed directly. Although DOT1L is the sole H3K79 methyltransferase, the dilution kinetics of H3K79me2 can vary depending on the proliferation rate. Quantifying the H3K79me2 signal in bone marrow-derived cell culture samples would clarify whether the deletion window allowed complete loss of the methylation mark.

3. Several observations are not discussed in sufficient depth:

- The finding that Dot1l deletion increases antigen-presentation signatures might reflect stress or activation rather than lineage fate change.

- The authors could also acknowledge that DOT1L's effect might be indirect, acting through cytokine feedback loops or altered progenitor proliferation, especially given the co-expression of Kit, Flt3, and Irf8 in early DC progenitors.

- Moreover, because H3K79 methylation is primarily associated with transcriptional elongation rather than initiation, the observed transcriptional changes could result from broader alterations in chromatin accessibility or polymerase processivity, rather than direct promoter regulation. Discussing this mechanistic aspect would help clarify whether DOT1L's role in DC differentiation reflects a direct control of lineage-defining gene expression or a secondary consequence of disrupted transcriptional elongation dynamics.

****Minor Comments****

1. Terminology:

The manuscript repeatedly refers to "mature" DCs-please clarify whether this means activated or fully differentiated cells.

2. Ontogeny statements:

The assertion that DCs of lymphoid origin are well established should be softened; the lymphoid contribution to some DC lineages remains under discussion.

3. Transitional DCs (tDCs):

The equivalence between tDCs and pre-cDC2As remains controversial. This should be acknowledged.

4. Cytokine supplementation:

The inclusion of SCF in the FLT3L-based differentiation assays should be justified, it is not a standard procedure.

5. Macrophage contamination:

The presence of C1qa, C1qb, and C1qc transcripts in some datasets suggests possible macrophage contamination. Please discuss how this was controlled for or how it might affect interpretation.

2. Significance:

Significance (Required)

This study provides important insights into the epigenetic regulation of DC differentiation by DOT1L. The conclusions would be more compelling if supported by in vivo validation or, alternatively, if the limitations of the current in vivo data were transparently acknowledged and the focus shifted toward mechanistic in vitro depth.

With these revisions, the manuscript would represent a valuable contribution to understanding how chromatin modification integrates with transcriptional control in shaping dendritic cell fate.

3. How much time do you estimate the authors will need to complete the suggested revisions:

Estimated time to Complete Revisions (Required)

(Decision Recommendation)

More than 6 months

4. Review Commons values the work of reviewers and encourages them to get credit for their work. Select 'Yes' below to register your reviewing activity at Web of Science Reviewer Recognition Service (formerly Publons); note that the content of your review will not be visible on Web of Science.

Yes

1. General Statements

We would like to thank the reviewers for their thoughtful and constructive comments on our manuscript. The reviewers make several suggestions that would strengthen our study. The major comment, shared by all three reviewers, is that our study would benefit from further clarifications and/or experimental validations regarding the differences observed between the *in vitro* and *in vivo* experiments. We carefully considered performing a mixed bone marrow chimera experiment. While we clearly see the value of the experiment, we had to conclude that at this moment it will not be possible for us to implement it. The two first authors are PhD students that have finished or are about to finish their thesis, which limits what we can do in terms of additional experiments, especially generating and analyzing mixed bone marrow chimeras, which will take 4-5 months after approval by the animal ethics committee. However, we do agree with the reviewers that the apparent discrepancy between *in vitro* and *in vivo* should be addressed more stringently. To do so, we made several substantial changes in the text following the suggestions of the reviewers and we included additional experimental results and analyses (e.g. Supplemental Figures 1A and 5E). Below we describe these changes in more detail in the point-by-point response.

Reviewer #1 (Evidence, reproducibility and clarity):

Summary:

In this study, Bouma et al. investigate the epigenetic mechanisms involved in dendritic cell (DC) development, focusing on the role of the lysine methyltransferase DOT1L, which mediates histone H3 lysine 79 (H3K79) methylation. The authors first show that *Dot1l* is expressed across most DC subsets and their progenitors. Consistently, DOT1L activity was detected in these subsets, as ChIP-seq analysis revealed an enrichment of H3K79 methylation marks around the transcription start sites of numerous genes that regulate DC fate. These marks were associated with active transcription, as confirmed by RNA sequencing. To assess the functional role of *Dot1l* in DC development, the authors used *Rosa26Cre-ERT2* × *Dot1l^{flox/flox}* mice. Bone marrow (BM) cells from these mice were treated *in vitro* with tamoxifen and cultured with FLT3L and SCF to induce DC differentiation. *Dot1l* deletion impaired the development of plasmacytoid DCs (pDCs) and enhanced the generation of conventional DC2 (cDC2), while leaving cDC1 development unaffected. Similarly, *in vivo* tamoxifen treatment of *Rosa26Cre-ERT2* × *Dot1l^{flox/flox}* mice for three days led to a comparable impairment of DC development upon *in vitro* culture of BM cells. Beyond mature DCs, *Dot1l* deletion also disrupted the ability of BM cells to generate common myeloid progenitors (CMPs), monocyte-dendritic cell progenitors (MDPs), and common DC progenitors (CDPs). These effects were attributed to the methyltransferase activity of DOT1L, as pharmacological inhibition of DOT1L produced similar outcomes. Interestingly, while *in vivo* tamoxifen treatment altered the frequencies of progenitor populations (MDP, CDP, CMP) in the BM, it did not significantly change the frequency of pDCs in the BM or spleen. Moreover, an increase in the cDC2 population was observed only in the BM, with no effect detected in the spleen. With these findings the authors claim that epigenetic regulation of gene expression by DOT1L is important for proper dendritic cell development.

Major comments.

While this study demonstrates that DOT1L regulates DC development *in vitro*, its inducible deletion *in vivo* using tamoxifen does not appear to significantly affect the overall distribution or function of DCs. Therefore, further investigation is needed to clarify the role of DOT1L in regulating DC fate under physiological conditions. The authors analyzed DC populations at only two time points (3 and 12 days) following tamoxifen-induced *Dot1l* deletion. As noted in the discussion, these time points are relatively early considering the lifespan of DCs, which often extends beyond this period. It would thus be important to assess the effects of *Dot1l* deletion over a longer duration (e.g., at least one month) to fully evaluate its impact on DC development. In addition to the BM, an extensive analysis of DCs population should be carried in the spleen as well as lymph nodes. Given the broad activity of the Rosa26-Cre system, prolonged deletion may affect overall mouse health and/or the function of other cell types that contribute to DC development; therefore, using a DC-specific Cre driver (e.g., CD11c-Cre) would provide a more targeted approach. Alternatively, competitive BM chimera experiments could be performed by reconstituting irradiated control mice with a 1:1 mixture of BM cells from Rosa26Cre-ERT2 × *Dot1l*^{flox/flox} and Rosa26Cre-ERT2 × *Dot1l*^{wt/flox} mice, both pre-treated with tamoxifen *in vitro*. Such experiments would offer more definitive evidence for the role of DOT1L in DC development *in vivo*.

>> Thank you for these suggestions. While we do agree that further *in vivo* validation, either by introducing a DC-specific Cre driver, or by generating mixed-bone marrow chimeras, could further strengthen the results obtained *in vitro*, these experiments will unfortunately not be feasible due to limitations in the personnel available. We have therefore opted to instead further clarify the discrepancies observed between *in vitro* and *in vivo* in the discussion of the manuscript, see page 21 of the discussion.

Aside from this point, the data and methods are clearly presented, and the figures are largely self-explanatory. All experiments were adequately replicated three times. Statistical analyses were primarily performed using t-tests, and ANOVA with multiple comparisons when appropriate. Since these are parametric tests that assume a normal distribution, it would be important to confirm whether the analyzed samples meet this assumption. If not, non-parametric tests should be used instead.

>> We checked for every ANOVA or t-test performed whether the residuals follow Gaussian distribution via QQ plots and confirmed that this is indeed the case.

Minor comments.

It would be informative to show how specific *Dot1l* expression is in DCs and their progenitors compared with other immune lineages (e.g., lymphocytes) and their precursors. The data suggest that DOT1L regulates H3K79 methylation of both shared and subset-specific genes among DC populations. The authors could elaborate on how this regulation achieves cell-type specificity—perhaps through differential *Dot1l* expression levels across DC subsets.

>> To compare *Dot1l* expression in DCs and their precursors to other immune cell types, we analyzed the publicly available RNAseq data set from Yoshida et al (2019) (doi.org/10.1016/j.cell.2018.12.036). This analysis showed that *Dot1l* is generally expressed in all immune cells and that *Dot1l* expression in DCs and their precursors clusters around the median of all immune subtypes analyzed. This analysis is now shown in new Supplementary Figure S1A. This finding agrees with other observations that cell-type specific roles of DOT1L are generally not related to DOT1L expression levels. For example, DOT1L is a dependency in MLL-rearranged leukemia but is not highly expressed in these cancer cells. Similarly, DOT1L safeguards CD8 T cell differentiation but is not highly expressed in CD8 T cells. Moreover, H3K79me1/2 are present

at similar levels in many different cell types, indicating that global DOT1L activity is not differentially regulated between cell types. We briefly discuss this and provide the relevant references on page 20.

Interestingly, Dot1l deletion both in vitro and in vivo markedly reduces the frequency of common DC progenitors (CDPs), which give rise to cDC1 and cDC2. The authors should discuss how such a substantial loss of progenitors does not proportionally affect downstream cDC populations.

>> We do not have a definitive answer to this valid question, but we now discuss the discrepancy in more detail in the text. A loss of CDPs could either be due to a decrease in differentiation towards CDPs, or by an accelerated differentiation from CDP towards more terminally differentiated cell-types. We elaborate on this further on major comment #1 from reviewer #2 and in the main text on page 14.

Although in vivo tamoxifen-induced deletion of Dot1l in Rosa26Cre-ERT2 × Dot1l^{flox/flox} mice does not significantly alter the overall distribution of DC subsets (pDCs and cDCs), it appears to modify their phenotype. It would therefore be valuable to examine how Dot1l loss impacts the functional properties of individual DC subsets. While pDC responsiveness to CpG stimulation seems preserved in the absence of Dot1l, assessing how cDCs respond to TLR3 and TLR4 stimulation and their capacity to activate T cells would provide important additional insights.

>> Unfortunately, we feel that isolating and functionally studying cDCs is currently outside of the main focus of the manuscript, which mainly looks at the loss of differentiation towards, and functionality of, pDCs as a consequence of DOT1L/H3K79me loss. Therefore, while we do agree with the reviewer that these assays could provide meaningful additional insights, we feel that this is unfortunately not feasible within the scope of this manuscript.

Reviewer #1 (Significance):

General assessment: Bouma et al. present compelling evidence that DOT1L is an important regulator of DC differentiation in vitro from bone marrow-derived cells. They further demonstrate that DOT1L regulates DC development through its lysine methyltransferase activity, mediating histone H3K79 methylation. While these in vitro findings are robust and well supported, the physiological relevance of DOT1L function in vivo remains less clearly established. Additional experiments would help to strengthen the conclusions regarding its role under physiological conditions.

Advance: While numerous transcription factors have been described as key regulators of DC subset development and fate, the role of epigenetic regulation in this process remains relatively understudied and poorly understood. This study addresses this important gap in the literature and provides novel insights into the role of H3K79 methylation mediated by DOT1L in controlling DC development.

Audience: This paper will be of interest for a specialized audience in the field of the regulation of dendritic cell ontogeny. This work could influence additional research to investigate the epigenetic regulation of DCs development.

Reviewer #2 (Evidence, reproducibility and clarity):

Bouma et al. present a comprehensive analysis of DOT1L-mediated histone H3K79 methylation across canonical DC subsets. By mapping the methylation landscape, the authors demonstrate that DOT1L regulates both shared and subset-specific gene programs. They show that in vitro or in vivo deletion of *Dot1l*, followed by in vitro differentiation, results in reduced myeloid progenitors and pDCs alongside an increase in cDC2s, while cDC1 numbers remain largely unaffected. Functionally, *Dot1l*-deficient DCs fail to produce IFN α upon stimulation. Transcriptomic profiling reveals enrichment of antigen presentation pathways in *Dot1l*-KO subsets, with upregulated MHC class II surface expression in pDCs. Mechanistically, pharmacological inhibition of DOT1L links these effects to its methyltransferase activity. Collectively, the data suggest that DOT1L differentially regulates canonical DC subset development and represses antigen presentation pathways.

The manuscript is well-written and technically sound. However, several conclusions would benefit from deeper discussion or additional experimental validation.

Major Comments

1. Interpretation of DC balance changes and cell-cycle effects

The authors propose that DOT1L loss skews DC differentiation toward a pDC-like phenotype. However, DOT1L deletion or inhibition, and the consequent global loss of H3K79 methylation, is well known to downregulate key cell-cycle genes (e.g., Cyclin D1, Cyclin E, CDK4/6, MCM family) while upregulating cell-cycle inhibitors (e.g., *Cdkn1a* and *b*). These transcriptional changes are associated with slower proliferation, G1 arrest or delayed S-phase entry, and reduced DNA replication fork progression. Importantly, blocking DNA synthesis (e.g., with aphidicolin or mitomycin C) during early culture inhibits DC emergence, underscoring that proliferation is essential for differentiation. The authors should discuss how their findings align with this established literature. Could the observed DC subset shifts result from impaired cell-cycle progression rather than lineage-specific transcriptional reprogramming? A more detailed consideration of this point is needed.

>> Thank you for the suggestions. If *Dot1l* KO induces a DC subset-specific decrease in cell proliferation, this could indeed cause a shift in subsets (i.e. we observed less pDC, more cDC2, unchanged cDC1). To address this possibility, we analyzed the transcript levels of the key cell-cycle related genes listed by the reviewer, as shown in the heat map on the right.

While loss of DOT1L induces differences in expression levels of the cell-cycle genes mentioned, the overall changes did not correlate with the shift in subsets. We now elaborate on the relationship between DOT1L and cell-cycle progression more extensively in the discussion on pages 20-21. Moreover, we also quantified the proliferation marker Ki67 by flow cytometry and observed that Ki67 increased in abundance in most of the identified cell-types present in BM and spleen, including the pDC, cDC1 and cDC2 (see new Supplemental Figure S5E and page 16 of the results section and pages 20-21 of the discussion). Therefore, a decrease in cell proliferation is likely not the main cause of the loss of differentiated pDCs. These data fit with our previous studies in which we show that the role of DOT1L in cell cycle progression is context dependent (e.g. see our findings in CD8

T cells in Malik et al 2025, Science Advances versus those in B cells in Göbel et al 2025, Blood). In general, while some cells require DOT1L for cell cycle progression, proliferation of other cell types is not affected by loss of DOT1L.

2. Discrepancy between in vitro and in vivo pDC phenotypes

The in vitro data show a marked reduction in pDCs, yet in vivo pDC numbers appear unchanged. Although the discussion briefly mentions proliferation differences, this discrepancy deserves a clearer explanation or experimental follow-up.

>> We agree with the reviewer that the differences observed between the *in vitro* and *in vivo* experiments indeed warrant further explanation. As described in our answer above, our results suggest that defects in cell proliferation are most likely not directly responsible for the pDC defect observed *in vitro*. As explained above, we now discuss the discrepancy between in vitro and in vivo in more detail on page 21.

Minor Comments

- Clarify statistical methods, specify biological replicate numbers, and indicate whether corrections for multiple comparisons were applied to transcriptomic analyses.

>> We provide all the information on the number of replicates and statistical methods used in the methods section and figure legends. We have clarified the use of multiple comparison corrections for the pathway analyses used in the 'Statistical analysis' section on page 11.

- The introduction is somewhat lengthy and repetitive; condensing it would improve focus.

>> To improve the focus and readability, we shortened the introduction significantly. Thank you for the suggestion.

- In the discussion sometimes it is not clear the distinction between findings and speculation.

>> We made several changes in the discussion to clarify the distinction between findings and speculation.

- Ensure consistent gene name formatting throughout (e.g., Dot1l, Dot1L).

>> We checked the formatting and made the appropriate changes.

Reviewer #2 (Significance):

The current manuscript fills a gap in knowledge, and this is its major strength. Other strengths are clarity and technical appropriateness.

The major weakness is that the work is mainly descriptive. Mechanistic insights into DOT1L-dependent transcriptional regulation are still weak.

The proposed mechanism -that DOT1L maintains pDC identity through H3K79 methylation at key transcription factors (Tcf4, SpiB, Irf8)- is intriguing but currently lacks functional evidence. The authors should consider validating this model experimentally, by modulating the expression of these genes without affecting DOT1L activity.

>> Thank you for the suggestion. Understanding the relevance of the candidate downstream targets of DOT1L in the phenotypes that we observed is indeed a major question. In support of the proposed mechanism, a body of literature already provides important insights into the role of the four transcription factors that we highlight in the manuscript. Deficiencies of Tcf4, SpiB, Bcl11a and IRF8 have all independently been demonstrated to result in lack of pDC development and function. To clarify this point, we modified the discussion in the manuscript and added extra references to support the proposed models (page 20).

Also the model suggesting that DOT1L indirectly represses antigen presentation via the Fbxo11-Ciita pathway is interesting but remains speculative. Additional mechanistic data would help support this claim.

>> We agree with the reviewer that the Fbxo11-Ciita pathway is interesting, but also speculative. We have adapted the text of the discussion accordingly (page 21).

Reviewer #3 (Evidence, reproducibility and clarity):

This manuscript investigates the role of DOT1L and its H3K79 methyltransferase activity in dendritic cell (DC) differentiation. The authors employ a combination of in vitro FLT3L/SCF bone marrow culture systems, in vivo inducible knockout models, and genome-wide H3K79me2 ChIP-seq and RNA-seq analyses to demonstrate that DOT1L influences the balance between pDC and cDC2 differentiation, while leaving cDC1 development largely unaffected. The study further identifies transcriptional and epigenetic programs associated with these changes, linking DOT1L deficiency to altered antigen presentation pathways and loss of pDC-associated transcription factors.

The paper provides valuable insights into DC biology. However, some of the key conclusions rely heavily on in vitro systems and short-term tamoxifen deletion models, which limit the interpretation of the in vivo data. Strengthening or clearly defining these limitations would substantially improve the paper's impact and clarity.

Major Comments

1. To strengthen the paper, the authors could follow one of two alternative strategies:
 - (1) Validate their in vitro observations through in vivo experiments, or
 - (2) Focus on deepening and refining their in vitro findings, moving the limited in vivo data to the supplementary material and explicitly acknowledging the limitations of the tamoxifen-inducible system.

Strategy 1 - Strengthen in vivo validation

- The experiments presented in Figures 3 and 5 could be repeated in a competitive bone marrow chimera setting (e.g. CD45.1/CD45.2 irradiated hosts reconstituted with a 1:1 mix of WT CD45.1⁺ and Dot1l-KO CD45.2⁺ cells).
- This design would allow dissection of direct (cell-intrinsic) versus indirect effects of DOT1L deficiency and could mitigate confounding effects of incomplete or asynchronous deletion.
- After reconstitution, mice could be maintained on tamoxifen-supplemented chow for a longer period to ensure efficient recombination and adequate time for observing phenotypic consequences.
- Flow cytometric analysis of spleen and bone marrow should use more refined panels to explore DC precursor and subset deficiencies. Suggested reference panels: Rodrigues et al., Immunity 2024; Minutti et al., Nat. Immunol. 2024; Zhu et al., Nat. Immunol. 2015.

Strategy 2 - Refine in vitro system and reposition in vivo data

- The authors could replicate their differentiation assays under conditions that emulate the chimera approach by co-culturing WT (CD45.1⁺) and Dot1l-KO (CD45.2⁺) bone marrow cells.
- This would reveal potential competition or cross-talk between WT and mutant cells and provide clearer mechanistic insight into cell-intrinsic versus extrinsic effects.
- The authors should examine how tamoxifen itself affects differentiation and measure the kinetics of deletion and H3K79me loss to better contextualize the dynamic response.
- It would also be valuable to assess which cDC2 subtypes (A vs. B) are preferentially affected by

Dot1l deficiency, again using more sophisticated flow cytometry panels (see references above). If this *in vitro*-focused strategy is adopted, the *in vivo* data could be moved to the supplementary material, with explicit acknowledgment that the inducible deletion model and the gradual nature of H3K79me dilution limit the interpretation of the *in vivo* findings.

>> Thank you for the outlined strategies. As described above for Reviewer #1, we agree that our manuscript could benefit from more extensive *in vivo* or *in vitro* validation, including the use of more sophisticated flow cytometry panels. However, given the limited time and resources currently available, we have expanded the discussion about the current limitations, both relating to the short time-frame used for *in vivo* studies, as well as the limited scope of our flow cytometry panels (e.g. see pages 20-21).

2. In Figures 2 and 3, the efficiency of H3K79me₂ depletion following Dot1l excision should be assessed directly. Although DOT1L is the sole H3K79 methyltransferase, the dilution kinetics of H3K79me₂ can vary depending on the proliferation rate. Quantifying the H3K79me₂ signal in bone marrow-derived cell culture samples would clarify whether the deletion window allowed complete loss of the methylation mark.

>> Unfortunately, in our experience, H3K79me₂ staining cannot be combined with the relevant surface marker stainings to define specific DC subsets and precursors due to the harshness of the H3K79me₂ staining procedure, which requires treatment with SDS to denature the nucleosomes and give access to H3K79.

3. Several observations are not discussed in sufficient depth:

- The finding that Dot1l deletion increases antigen-presentation signatures might reflect stress or activation rather than lineage fate change.

>> We indeed detect broad pro-inflammatory effects in the DOT1LKO background irrespective of the DC subset analyzed (Figure 5). In the experiments described in the manuscript, we cannot determine whether the increased MHC class II expression is cell-intrinsic or due to extrinsic inflammatory signals, such as cytokines. We now discuss these findings and their limitations more carefully on pages 21-22.

- The authors could also acknowledge that DOT1L's effect might be indirect, acting through cytokine feedback loops or altered progenitor proliferation, especially given the co-expression of Kit, Flt3, and Irf8 in early DC progenitors.

>> We can indeed not exclude cell-extrinsic effects of DOT1L mediated e.g. via cytokine feedback loops. In addition, the effects of *Dot1l*-KO are present at multiple progenitor stages as we observed significant decreases in BM precursors. Given the results with the Ki67 staining as discussed above, we believe that the changes observed are unlikely to be caused by a general reduction in proliferation in *Dot1l*-KO bone marrow. As described above, we discuss this in more detail on pages 20-22 of the discussion.

- Moreover, because H3K79 methylation is primarily associated with transcriptional elongation rather than initiation, the observed transcriptional changes could result from broader alterations in chromatin accessibility or polymerase processivity, rather than direct promoter regulation. Discussing this mechanistic aspect would help clarify whether DOT1L's role in DC differentiation reflects a direct control of lineage-defining gene expression or a secondary consequence of disrupted transcriptional elongation dynamics.

>> The reviewer points at one of the main unanswered questions related to DOT1L, i.e. how H3K79 methylation, which predominantly occurs in transcribed gene bodies, might affect gene regulation. H3K79me₂ is a co-transcriptional histone modification associated with active transcription and the level of H3K79me₂ typically correlates with the level of transcription.

However, if and how H3K79me2 in turn affects gene transcription is still poorly, if at all, understood. While most observations, including from our team, suggest that H3K79me2 promotes transcription, a recent study proposes a negative role in transcription elongation (<https://doi.org/10.21203/rs.3.rs-3055610/v1>). Other studies have obtained evidence, for roles of DOT1L independent of its catalytic activity and we made similar observations for Dot1 in yeast. One major challenge in linking H3K79me to gene regulation is the slow turnover of the modification, which severely complicates identifying the direct targets of DOT1L that require H3K79me2 for proper gene regulation. We now provide a short summary of the standing of the field in the discussion to acknowledge the open questions and other possible mechanisms on page 22.

Minor Comments

1. Terminology: The manuscript repeatedly refers to "mature" DCs-please clarify whether this means activated or fully differentiated cells.

>> We thank the reviewer for pointing out the confusing terminology. We changed "mature" to "differentiated" or "fully differentiated" in the manuscript where applicable to make the distinction between differentiation and TLR-induced maturation clearer.

2. Ontogeny statements: The assertion that DCs of lymphoid origin are well established should be softened; the lymphoid contribution to some DC lineages remains under discussion.

>> We agree with the reviewer that this is a field of active research and we have worded this more carefully to emphasize that these data are from recent papers that should be further confirmed.

3. Transitional DCs (tDCs): The equivalence between tDCs and pre-cDC2As remains controversial. This should be acknowledged.

>> We acknowledge that the relation between tDCs and pre-cDC2 is controversial and have adapted the text of the manuscript accordingly.

4. Cytokine supplementation: The inclusion of SCF in the FLT3L-based differentiation assays should be justified, it is not a standard procedure.

>> While SCF supplementation is not included in the standard procedure for FLT3L bone marrow cultures, it has been described to specifically expand hematopoietic stem cells and early DC progenitors to increase DC yields (Ou EJI 2023, DOI: 10.1002/eji.202250201). We have added this reference to the manuscript.

5. Macrophage contamination: The presence of C1qa, C1qb, and C1qc transcripts in some datasets suggests possible macrophage contamination. Please discuss how this was controlled for or how it might affect interpretation.

>> It is indeed known that macrophage blebs can be attached to other cell types purified by sorting as has, for example, been described by Millard et al (Cell reports 2021, <https://doi.org/10.1016/j.celrep.2021.110058>). To investigate this possibility, we checked for changes in macrophage specific markers Adgre1, Siglec-1, Mertk and Vcam between the KO and WT DC subsets (Figure to the right). We indeed detect the majority of these macrophage markers at elevated levels in DOT1L-KO pDCs and cDC2s. However, in cDC1s, these genes were either undetected, unaltered, or only modestly changes in Dot1l-KO. In contrast, C1q transcripts were homogenously upregulated in all sorted KO subsets. Therefore, while we cannot fully exclude contamination, we

Full Revision

believe that the changes in C1q transcripts observed cannot solely be due to macrophage contamination. We now briefly discuss this on page 17.

Reviewer #3 (Significance):

This study provides important insights into the epigenetic regulation of DC differentiation by DOT1L. The conclusions would be more compelling if supported by in vivo validation or, alternatively, if the limitations of the current in vivo data were transparently acknowledged and the focus shifted toward mechanistic in vitro depth.

With these revisions, the manuscript would represent a valuable contribution to understanding how chromatin modification integrates with transcriptional control in shaping dendritic cell fate.

February 18, 2026

RE: Life Science Alliance Manuscript #LSA-2026-03624

Dr. Fred van Leeuwen
Netherlands Cancer Institute
Gene Regulation, B4
Plesmanlaan 121
Amsterdam 1066CX
Netherlands

Dear Dr. van Leeuwen,

Thank you for submitting your revised manuscript entitled "Histone methyltransferase DOT1L differentially affects the development of dendritic cell subsets". As you will see, both original reviewers are overall satisfied with the revisions and note the limitations in time and personnel that prevented further experimental validation of these observations in vivo. While the reviewers have no further requests, we concur with Reviewer 2 that the conflicting in vitro vs. in vivo observations (nicely documented in the discussion) should be indicated more clearly to the reader. A suitably revised manuscript must modify the sentence in lines 44-46 in the abstract to indicate that discordant results were found. We would be happy to publish your paper in Life Science Alliance pending this change and final revisions necessary to meet our formatting guidelines.

MANUSCRIPT ORGANIZATION AND FORMATTING:

To avoid unnecessary delays in the acceptance and publication of your paper, please read the following information carefully. Full guidelines are available on our Instructions for Authors page, <https://www.life-science-alliance.org/authors>

- Please upload your main and supplementary figures as single files.
- Please add a Running Title and a Summary Blurb/Alternate Abstract in our system
- Please add a Category for your manuscript in our system.
- Please add the X and Bluesky handles of your host institute/organization, as well as your own and/or one of the authors, in our system.
- Please upload a clean version of the manuscript file without the tracking changes. The version that highlights changes made to your manuscript file should be uploaded with the file designation "Related Manuscript file."
- Please mark the secondary Corresponding Author in our system as well.
- Please consult our manuscript preparation guidelines <https://www.life-science-alliance.org/manuscript-prep> and make sure your manuscript sections are in the correct order.
- Please add a Data Availability statement to your manuscript.
- Please rename "Competing interests" to "Conflict of Interest."
- Please add the authors' contributions to our system.
- Please add your main, supplementary figure, and table legends to the main manuscript text after. It is recommended to exclude figures from the manuscript text and upload them separately.
- Please use the [10 author names et al.] format in your references (i.e., limit the author names to the first 10).
- Please add molecular weight markers to the blots in Figure S3C.

LSA encourages authors to provide a 30-60 second video where the study is briefly explained. We will use these videos on social media to promote the published paper and the presenting author (for examples, see

<https://docs.google.com/document/d/1-UWCfbE4pGcDdcgzcmiuJl2XMBJnxKYeqRvLLrLSo8s/edit?usp=sharing>). Corresponding or first-authors are welcome to submit the video. Please submit only one video per manuscript. The video can be emailed to contact@life-science-alliance.org

FINAL FILES:

The following items are required for acceptance.

The license to publish form must be signed before your manuscript can be sent to production. A link to the license to publish form will be available to the corresponding author only. Please take a moment to check your funder requirements.

Thank you for your attention to these final processing requirements. Please revise and format the manuscript and upload materials as soon as you are able.

Thank you for this interesting contribution to the literature. We look forward to publishing your paper in Life Science Alliance.

Sincerely,

Reviewer #1 (Comments to the Authors (Required)):

The authors have somewhat improved the discussion, but this paper remains descriptive; the authors have decided not to deepen the mechanistic understanding of Dot1L activity in DC differentiation. Moreover, whether there actually is a functional role of Dot1L activity in DC differentiation is even less clear than before: a few easy ways to explain the change in DC populations in Dot1L-depleted or inhibited cells have been excluded, and the authors now underline that Dot1L is broadly expressed in many blood cell lineages. The most significant red flag, the discordance between in vitro and in vivo results, remains.

Reviewer #2 (Comments to the Authors (Required)):

In this study, the authors present compelling in vitro evidence that DOT1L is an important regulator of dendritic cell (DC) differentiation from bone marrow-derived cells. They further demonstrate that DOT1L regulates DC development through its lysine methyltransferase activity, mediating histone H3K79 methylation. These in vitro findings are solid, well presented, and strongly supported by the data. However, the physiological relevance of DOT1L function in vivo remains insufficiently established. Together with the other reviewers, I previously encouraged the authors to perform additional experiments to strengthen their conclusions regarding the role of DOT1L in DCs under physiological in vivo conditions. The authors indicate that, due to limited personnel and resources, they were unable to address this major point experimentally. Nevertheless, they have responded carefully to my other comments and expanded the discussion to better contextualize their findings and to discuss certain experimental discrepancies in greater detail.

Despite this important limitation, the manuscript is of interest because it provides the first compelling evidence implicating DOT1L-mediated epigenetic regulation in DC development and function, thereby addressing an important gap in the literature. I therefore support publication of the manuscript, provided that the authors clearly outline the limitations of the current study and explicitly state that their observations require in vivo validation to further substantiate the role of DOT1L-mediated H3K79 methylation in the regulation of DC development.

March 3, 2026

RE: Life Science Alliance Manuscript #LSA-2026-03624R

Dr. Fred van Leeuwen
The Netherlands Cancer Institute
Gene Regulation, B4
Plesmanlaan 121
Amsterdam 1066CX
Netherlands

Dear Dr. van Leeuwen,

Thank you for submitting your Research Article entitled "Histone methyltransferase DOT1L differentially affects the development of dendritic cell subsets". It is a pleasure to let you know that your manuscript is now accepted for publication in Life Science Alliance. Congratulations on this interesting work.

DISTRIBUTION OF MATERIALS:

Again, congratulations on a very nice paper. I hope you are pleased with how the manuscript was handled editorially. We look forward to future exciting submissions from your lab.

Sincerely,
